# Look twice: A generalist computational model predicts return fixations across tasks and species

**Mengmi Zhang** [1,2,3◔], **Marcelo Armendariz** [1,2,4◔], **Will Xiao** [5], **Olivia Rose** [5], **Katarina Bendtz** [1,2], **Margaret Livingstone** [5], **Carlos Ponce** [5], **Gabriel Kreiman** [1,2]*

**1** Boston Children's Hospital, Harvard Medical School, Boston, Massachusetts, United States of America, **2** Center for Brains, Minds and Machines, Cambridge, Massachusetts, United States of America, **3** CFAR and I2R, Agency for Science, Technology and Research, Singapore, **4** Laboratory for Neuro- and Psychophysiology, KU Leuven, Leuven, Belgium, **5** Department of Neurobiology, Harvard Medical School, Boston, Massachusetts, United States of America

◔ These authors contributed equally to this work.

* gabriel.kreiman@tch.harvard.edu

**Data Availability Statement:** All the source code and raw data is publicly available through the lab's GitHub repository: https://github.com/kreimanlab/RefixationModel.

## Abstract

Primates constantly explore their surroundings via saccadic eye movements that bring different parts of an image into high resolution. In addition to exploring new regions in the visual field, primates also make frequent return fixations, revisiting previously foveated locations. We systematically studied a total of 44,328 return fixations out of 217,440 fixations. Return fixations were ubiquitous across different behavioral tasks, in monkeys and humans, both when subjects viewed static images and when subjects performed natural behaviors. Return fixations locations were consistent across subjects, tended to occur within short temporal offsets, and typically followed a 180-degree turn in saccadic direction. To understand the origin of return fixations, we propose a proof-of-principle, biologically-inspired and image-computable neural network model. The model combines five key modules: an image feature extractor, bottom-up saliency cues, task-relevant visual features, finite inhibition-of-return, and saccade size constraints. Even though there are no free parameters that are fine-tuned for each specific task, species, or condition, the model produces fixation sequences resembling the universal properties of return fixations. These results provide initial steps towards a mechanistic understanding of the trade-off between rapid foveal recognition and the need to scrutinize previous fixation locations.

## Author summary

We move our eyes several times a second, bringing the center of gaze into focus and high resolution. While we typically assume that we can rapidly recognize the contents at each fixation, it turns out that we often move our eyes back to previously visited locations. These return fixations are ubiquitous across different tasks, conditions, and across species. A computational model captures these eye movements and return fixations by using four key mechanisms: extraction of salient parts of an image, incorporation of task goals such

**Funding:** This work was supported by NIH grant R01EY026025 (GK), by NRF grant AISG2-RP-2021-025 (MZ), and by the Center for Brains, Minds and Machines, funded by NSF Science and Technology Centers Award CCF-1231216 (GK). MZ is supported by postdoctoral fellowship, CFAR Early Career Investigatorship, and Startup Fund C210415012 (MZ) of the Agency for Science, Technology and Research. MA is supported by a postdoctoral fellowship of the Research Foundation Flanders (FWO 1230521N). The funders had no role in study design, data collection and analysis, decision to publish, or preparation of the manuscript.

**Competing interests:** The authors have declared that no competing interests exist.

as the target during visual search, a constraint to avoid making large eye movements, and forgetful memory of previous locations. Neither the extreme of getting stuck at a single location or the extreme of never revisiting previous locations seems adequate for visual processing. Instead, the combination of these four mechanisms allows the visual system to achieve a happy medium during scene understanding.

## Introduction

Primates and other animals move their eyes several times a second through ballistic excursions called saccades, bringing different parts of a scene into high resolution at the center of fixation. Saccades are critical to visual processing and are orchestrated by multiple brain areas involved in determining the location of the next fixation and programming the corresponding motor commands [1–6]. Certain locations in an image are more salient in the sense that they draw more fixations; for example, subjects rarely make saccades to the middle of a white wall, but they will saccade to a moving yellow car.

Models that aim to predict eye movements generally postulate an attention map that specifies how saliency differs across an image (e.g., [7–16]) A winner-take-all mechanism selects the maximum of the attention map as the location for the next fixation. Thereafter, some change must occur in the attention map to allow the eyes to explore other locations and prevent the system from repeatedly selecting the same maximum. An inhibition-of-return (IOR) mechanism is typically imposed to ensure that the model can choose the next maximum [17]. A balance must be struck between an IOR mechanism that is too strong, which would prevent the system from scrutinizing the areas of maximum interest in the attention map, and a weak IOR, which would prevent image exploration.

A finite IOR mechanism would allow subjects to return to previously visited locations. Indeed, behavioral studies have shown that *return fixations* often take place during normal gaze behaviors including reading [18], pattern copying or block sorting [19, 20], portrait painting [21], solving arithmetic and geometry problems [22], visual search [17, 23–28], and free viewing [24, 29–31]. Return fixations have been used in neurophysiological studies of target detection [32], to study working memory during visual search in change detection tasks [17, 27, 28, 33], in object and location recall tasks [31], to study the effects of memory load [25, 31, 34–38], in rejection of distractors [24], and the comparison between IOR and memory-less models [39].

The neural mechanisms driving return fixations remain poorly understood and likely involve multiple factors including the content of the visual field, contextual relations among objects, goal-relevance and task instructions, eccentricity-dependent sampling, object familiarity, visual working memory, and eye muscle constraints [33, 39–43]. These different factors could change across different experimental conditions. Several studies examined the trade-offs between scrutinizing information at a given scene location and the drive to examine new salient locations during eye movements, *e.g.*, [7, 12, 44–47]. For example, the Bayesian models proposed in [7, 12, 46] generate saccades for scene viewing and fit human eye movement data. Recent work has also characterized strategies of deploying eye movements within image regions for exploitation or other regions for exploration during free-viewing of natural images by humans [43].

While previous studies focused only on a single task performed by one species under specific conditions, here we set out to quantify and model the *universal* properties of return fixations across a wide variety of naturalistic tasks, conditions, and across two species. We assessed

the general principles underlying return fixations, as opposed to only capturing how locations are revisited under a single experiment. We studied eight experiments that included different species (humans and monkeys), different types of images (isolated object arrays, natural images, and Waldo images), different tasks (free viewing and visual search), and different stimulus dynamics (static images and egocentric videos). We show that both monkeys and humans make frequent return fixations across all the tasks and conditions studied. These return fixations have few intervening saccades, often follow a 180-degree turn in saccadic direction, and are more prevalent in areas of high saliency and areas of high similarity to the target during visual search. To gain insight into the potential neural mechanisms that drive return fixations, we propose a proof-of-principle, image-computable, and biologically-inspired neural network model of eye movements. Without fine-tuning parameters for specific image types, tasks, species or condition, the model provides a first-order approximation to the spatiotemporal dynamics underlying revisiting of previous fixation locations. The key ingredients of the model leading to the universal properties of return fixations include an image feature extractor, bottom-up saliency, target similarity for visual search tasks, a finite inhibition-of-return memory, and a constraint on the saccade sizes. The ubiquitous and frequent nature of return fixations suggests that these model components can also provide important building blocks to build better neural network models of object recognition and visual search.

## Results

A typical eye movement sequence is shown in Fig 1A. The subject fixated at location 3 (red triangle), then made a saccade to location 4 (yellow circle), and quickly returned to location 5 (red circle), which overlaps with location 3. We refer to location 3 as a *to-be-revisited* fixation, location 5 as a *return fixation*, and to all other locations as *non-return fixations*. We used a threshold of one degree of visual angle (dva, approximately the resolution of our eye tracking system, Methods) to determine whether two fixations overlapped. Other examples of return fixations are shown in Fig 1B–1D. We evaluated the pattern of return fixations in eight experiments schematically illustrated in Fig 2 (see Methods for experiment details). These eight experiments encompassed different primate species (humans, Fig 2A–2D and 2G–2H, and non-human primates, Fig 2E–2F), different tasks (free viewing, Fig 2D–2G, and visual search, Fig 2A–2C and 2H), and different stimulus presentation formats (static images, Fig 2A–2F, and free-moving recorded in egocentric videos, Fig 2G–2H). We characterized the prevalence of return fixations and their properties, and propose a biologically-inspired computational model that captures the main properties of return fixations.

### Return fixations are ubiquitous

We started by re-examining the fixation patterns of human subjects during three progressively more challenging visual search tasks in a dataset that we had studied previously [8], which included object array images (Fig 2A), natural images (Fig 2B), and "Where is Waldo?" images (Fig 2C). Even though many computational models of visual search assume infinite inhibition of return (IOR)—that is, without possibility of returning to a previously visited location, we observed that humans made a large number of return fixations in all three cases: 11.8 ± 0.7% (Fig 3A, here and throughout, mean ± SEM across subjects), 18.8 ± 0.6% (Fig 3B), and 15.6 ± 0.9% (Fig 3C), respectively. In all cases, the proportion of return fixations was higher than expected by a null model implementing random eye movements while respecting the distribution of saccade sizes ($p < 10^{-5}$, two-tailed t-test, $df$ = 14, Methods); this was consistent with previous studies [17, 27, 28, 33]. Moreover, subjects returned to the same location not just

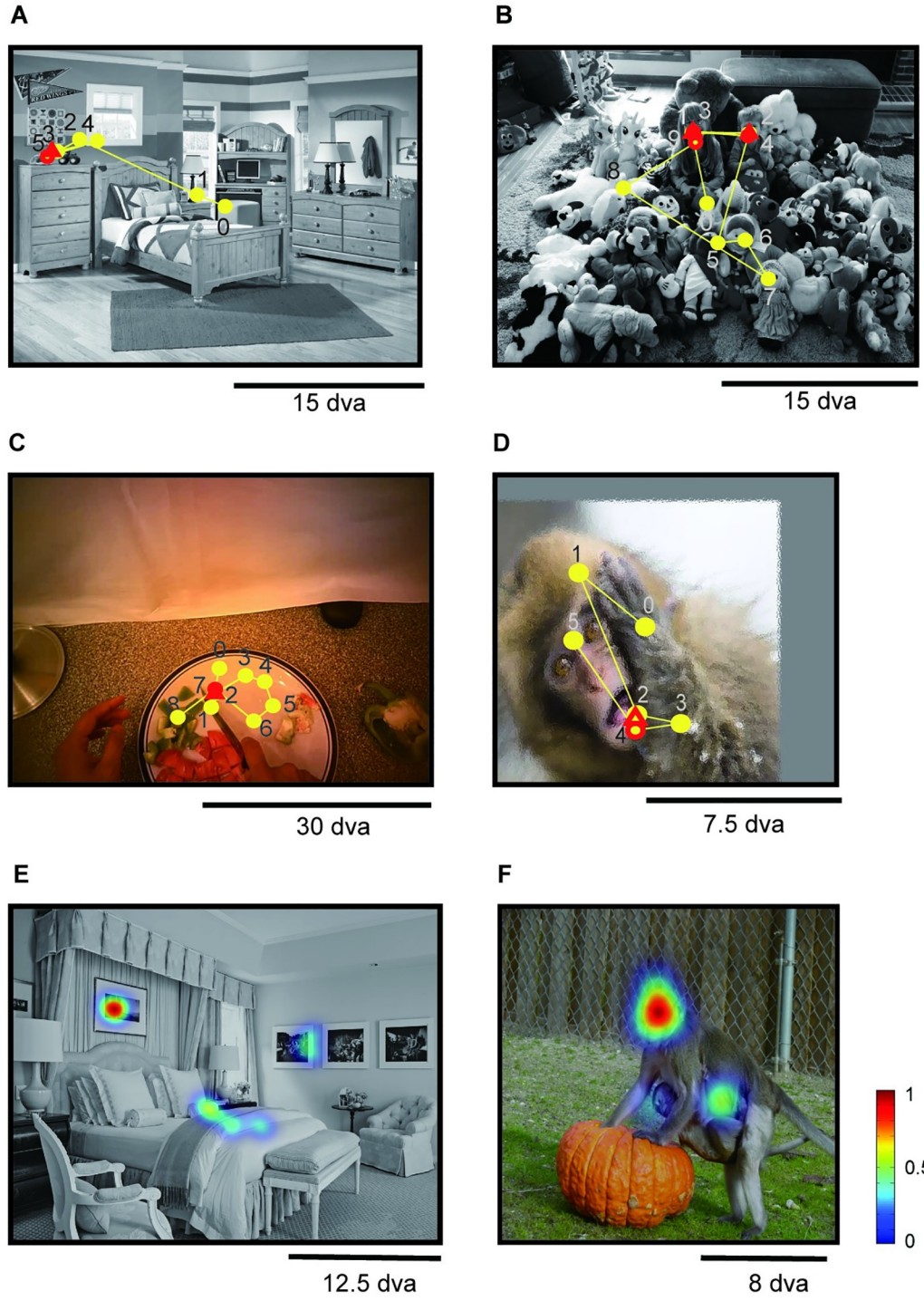

**Fig 1. Primates make return fixations during natural vision.** Example fixation sequences (yellow circles) of humans (**A-C**) and monkeys (**D**) during visual search (**A**), free viewing of static images (**B, D**), and a cooking task with a head-mounted eye tracker (**C**) (see Fig 2 for task definitions). The numbers denote the fixation order. A fixation (yellow circles) is referred to as a "return fixation" (red triangle) if the Euclidean distance to any of the previous fixations is less than 1 degree of visual angle (dva). The previous fixation overlapping with the return fixation is referred to as a "to-be-revisited" fixation (red circle). There are two return fixations in **B** and one in **A, C**, and **D**. **E-F** Return fixations are consistent across subjects. **E** and **F** show example images of consistent return fixations across subjects in free viewing tasks. Color bar on the right shows the scales of between- subject consistency (see S5 Fig for more examples). Photo sources: (A, B, E) were modified from a public dataset [8]. (C) was modified from a public dataset [48]. (D, F) Reproduced with permission from BPRC and pexels.com.

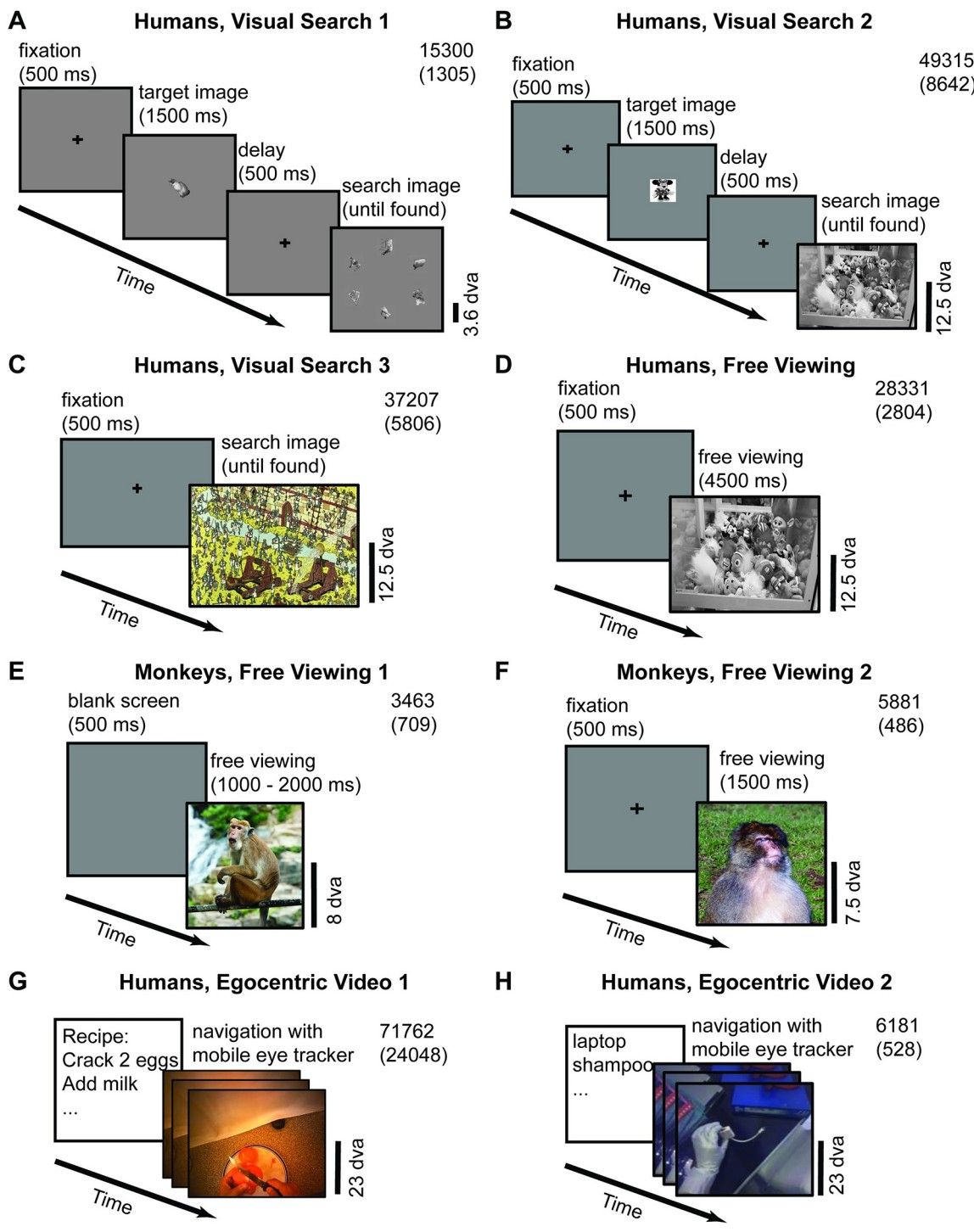

**Fig 2. Schematic description of the eight experimental paradigms.** (**A-C**) Visual search tasks with object arrays (**A**), natural images (**B**), and "Waldo" images (**C**) (see [8] for details). (**D-F**) Free viewing experiments with static natural images in humans (**D**) and monkeys (**E, F**). (**G**) Egocentric video dataset where subjects had to follow various recipes to make breakfast (see [48] for details). (**H**) Egocentric video dataset where subjects had to search for 22 items (see reference [49] for details). The numbers in the top right corner of each subplot denote the total number of fixations (top) and the total number of return fixations (bottom). Photo sources: (A, B, C, D) were modified from a public dataset [8]. (G) was modified from a public dataset [48]. (H) was modified from a public dataset [49]. (EF) Reproduced with permission from BPRC and pexels.com.

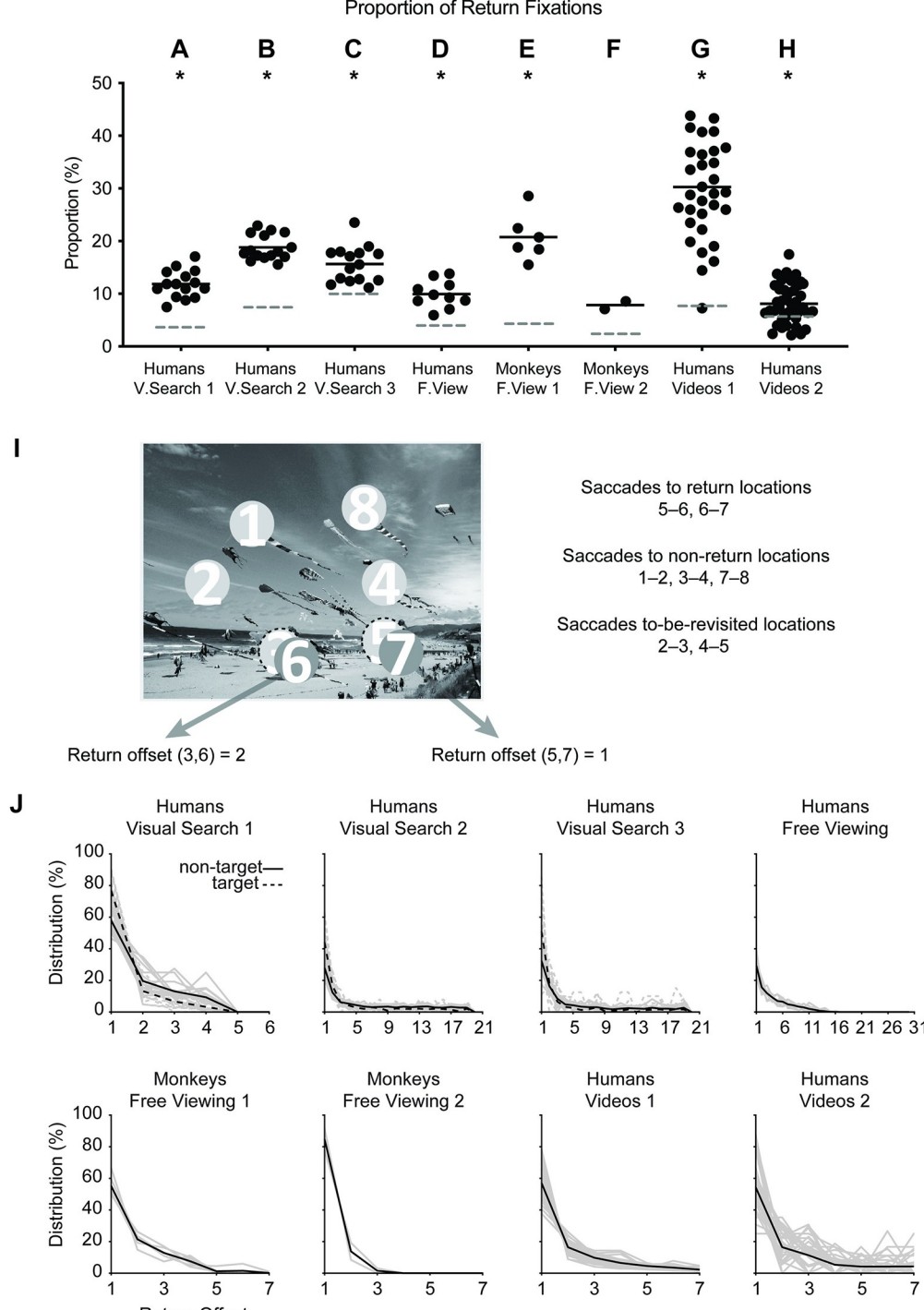

**Fig 3. Human and non-human primates make frequent return fixations.** (**A-H**) Proportion of return fixations, i.e., total number of return fixations normalized by total number of fixations, for each of the 8 experiments (Fig 2). Each dot indicates an individual subject (horizontal spread is only for visualization). The horizontal solid line shows the average across subjects. The chance level (dashed lines) was computed by generating sequences of random fixations (Methods). Asterisks indicate significant differences from chance ($p < 0.05$, one-sample t-test). We did not conduct a statistical test in the monkey free viewing 2 experiment as there were only 2 subjects. (**I**) Example sequence of 8 fixations on an image, including return fixations (6, 7), to-be-revisited fixations (3, 5), and non-return fixations (1, 2, 4, 8). The return offset is the number of intervening fixations for a given return location. (**J**) Distribution of the return offset for the 8 experiments. The light grey line shows each subject and the black lines show averages. In the visual search experiments, the solid and dashed lines show return fixations to non-targets and targets, respectively. Photo sources: (I) was modified from a public dataset [8].

once, but often multiple times. S1 Fig shows the proportion of cases where subjects made two return fixations.

We reasoned that return fixations could be particularly relevant during visual search, especially in difficult tasks, where it is easy to miss the target and it may therefore be advantageous to revisit previous locations [8]. To assess whether return fixations constitute a unique property of visual search tasks, we conducted a free-viewing experiment where there was no obvious incentive to revisit previous locations (Fig 2D). Under free-viewing conditions, subjects still made multiple return fixations (Fig 3D), 9.9 ± 0.8%, above the proportion expected by chance ($p < 10^{-5}$, two-tailed t-test, $df = 9$). The free-viewing experiment in Fig 2D used the same images as the visual search experiment in Fig 2B and we can therefore directly compare the fraction of return fixations. There were nearly twice as many return fixations during visual search compared to free-viewing conditions.

To evaluate whether return fixations are unique to humans, we analyzed data from two additional free-viewing experiments in macaque monkeys (Fig 2E–2F). Monkeys also demonstrated extensive return fixations: 20.7 ± 1.8% and 7.8 ± 1.0% (Fig 3E–3F). In the free-viewing experiment 1, the proportion of return fixations was higher than expected by chance ($p < 10^{-4}$, two-tailed t-test, $df = 5$). Consistently, the proportion of return fixations was also higher than chance for the two monkeys that participated in the free-viewing experiment 2 (we cannot report accurate statistics across individuals for this experiment with N = 2). The proportion of return fixations made by free-viewing monkeys was comparable or higher than the proportion by humans during visual search. It should be noted that image content, image sizes, and stimulus presentation times differed between the monkey and human experiments. Therefore, the different proportions of return fixations between humans and monkeys may reflect differences in stimulus conditions, rather than a true difference between species.

A large fraction of eye movement studies have focused on behavioral responses to flashed static images. Intrigued by the consistency of return fixations across visual search and free viewing of static images, we asked whether subjects also revisit fixation locations during more naturalistic conditions. To address this question, we extended the analyses to two egocentric video datasets where eye movements were tracked in free-moving subjects during a cooking task (Fig 2G, reference [48]), or a real-world visual search task (Fig 2H, reference [49]). In the cooking egocentric video dataset, subjects were asked to follow a sequence of steps on recipes to prepare a meal (Methods). In the visual search egocentric video dataset, subjects were asked to navigate an indoor home, search for a list of commonly used items, such as a thumb drive, and put those items on a designated table (Methods). To avoid the complexities of head movements and also to account for fixation locations that may disappear from the field of view, we focused on stable five-second segments (S2 Fig, Methods). Under these conditions, subjects still made repeated return fixations: 30.3 ± 1.8% (cooking) and 8.1 ± 0.5% (visual search task) (Fig 3G–3H). In both egocentric video datasets, the proportion of return fixations was higher than expected by chance ($p < 10^{-13}$, $df = 31$, for video dataset 1 and $p = 0.02$, $df = 43$, for video dataset 2). During the cooking task, subjects manipulated kitchenware, foods, and the recipe in front of them and tended to make a large number of return fixations. In sum, return fixations were ubiquitous across tasks, species, and static images or free-moving conditions.

The total number of fixations varied across tasks and the proportion of return fixations depends on the total number of fixations in a non-linear way. We therefore evaluated the prevalence of return fixations during the first six fixations S3 Fig. We chose to examine the first six fixations because this number allowed us to incorporate most of the data, including those experiments that had few fixations per trial. Except for the visual search 3 experiment, the proportion of return fixations was above chance in all the experiments, even when considering exclusively the first six fixations in each trial.

In sum, return fixations are ubiquitous and are not restricted to visual search tasks. Subjects revisit certain locations within an image and during free movement, even multiple times, across a large variety of experimental conditions, tasks, and species.

## Return fixations are consistent across subjects

We asked whether the locations of return fixations were consistent across subjects in the first six experiments with static images (Fig 1E–1F and S4 Fig). We omitted this analysis in the egocentric video datasets because the field of view in each frame could be different across subjects, making comparisons between subjects difficult to interpret. S5 Fig shows examples that illustrate consistent return fixation locations across subjects for the same image. For example, seven out of ten subjects made a return fixation to the location at "9 o'clock" in S5(A) Fig and five out of seven subjects made return fixations to the framed picture on the upper left in S5 (D) Fig. To quantify the degree of consistency across subjects for a given image, we divided the image into a grid and calculated the probability of observing return fixations at each location (Methods, S4(B) Fig). We summarized these probability distributions of return fixation locations by computing their entropy. An extreme case of perfect consistency would lead to a probability of 1 at a given location and 0 elsewhere, resulting in minimal entropy. In contrast, a complete lack of consistency would lead to an approximately uniform probability distribution, except for random overlaps, resulting in high entropy. The chance level was computed by considering the same total number of return fixations and distributing them at random locations in the image (Methods). The entropy was lower than expected by chance in four of the six experiments, the exceptions being the visual search 3 experiment and the Free Viewing 2 experiment in monkeys, both of which showed the same trend but did not reach statistical significance (S4(C) Fig). In sum, in most experiments, different subjects tended to revisit the same locations.

## Subjects revisited return-fixation locations and lingered at those locations

We divided all fixation locations into the following three non-overlapping categories: to-be-revisited fixations, non-return fixations, and return fixations (Fig 3I). We calculated the offset between to-be-revisited and return fixations. In the two examples in Fig 3I, the return offset between fixations 5 and 7 was 1 (intervening location 6) and the return offset between fixations 3 and 6 was 2 (intervening fixations 4 and 5). A strong inhibition-of-return would imply that there should be a large return offset between to-be-revisited and return fixations. In stark contrast, in all the experiments, the distribution of return offsets showed a rapid decay (Fig 3J). The percentage of all return fixations with an offset of 1 ranged from 29.2% (humans, Free Viewing) to 84.4% (monkeys, Free Viewing 2) and the percentage of return fixations with a return offset less than or equal to 3 ranged from 56.8% (humans, Free Viewing) to 100% (monkeys, Free Viewing 2). Regardless of the species, experimental task or stimulus mode, subjects tended to move their eyes back to previously visited locations after a very short delay, often the minimum possible delay of one intervening fixation.

After returning their gaze to a given location, subjects tended to fixate longer at the return location, especially during visual search (Fig 4A). The difference between the return fixation durations and non-return fixation durations ranged from $36.8 \pm 7$ ms (Human Visual Search 1) to $79.6 \pm 6$ ms (Human Visual Search 2). The duration of return fixations was significantly longer than non-return fixations for all the visual search experiments and two of the free-viewing experiments. Intriguingly, in the egocentric video visual search experiment, fixation durations were longer for the non-return fixations (see Discussion).

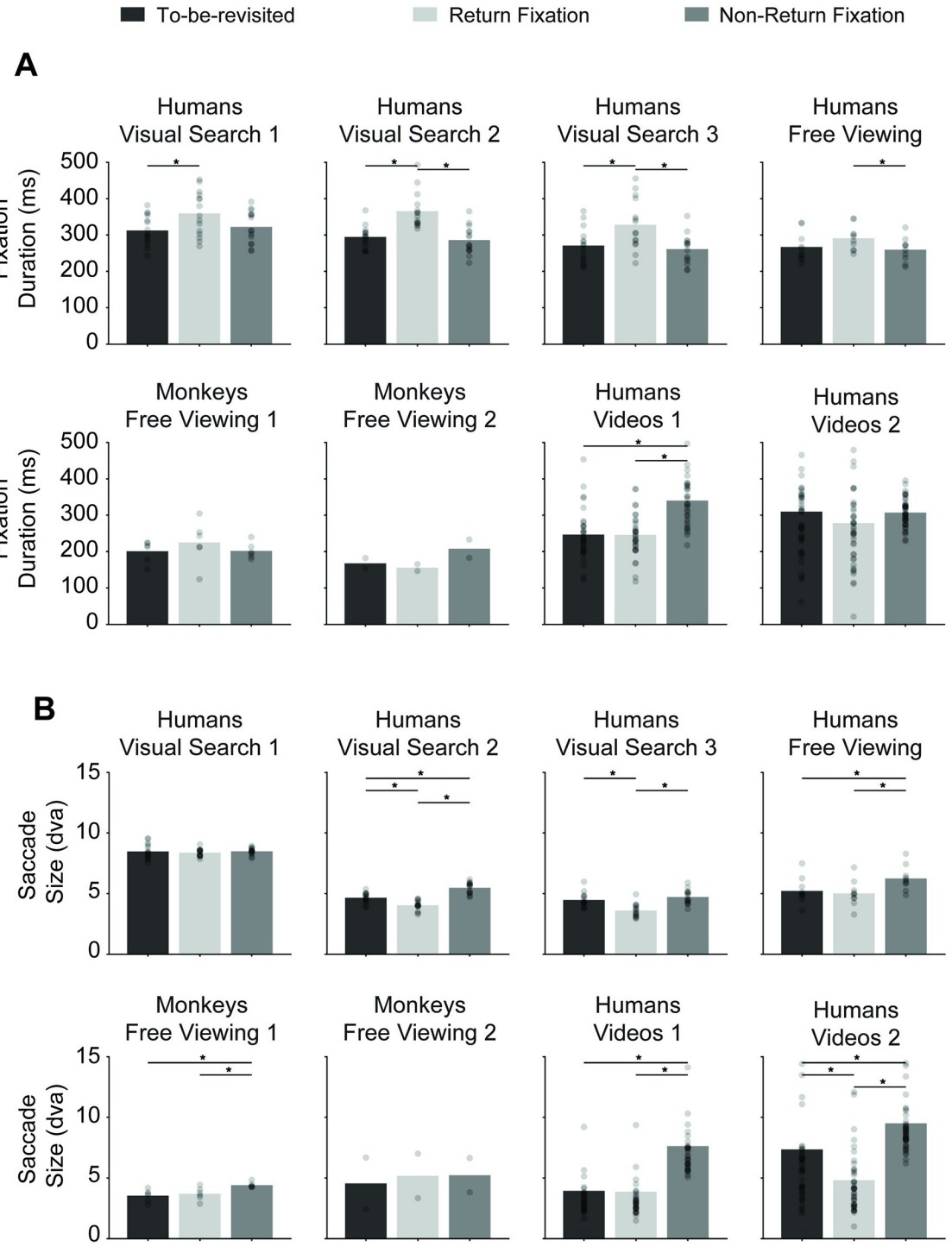

**Fig 4. Return fixations tended to last longer and follow smaller saccades.** (**A**) Duration of fixations in non-return fixation locations (dark gray), to-be-revisited locations (black), and return fixation locations (light gray). Dots show individual subjects. Error bars denote SEM. * denotes $p < 0.05$, two-tailed t-test. The durations of return fixations tended to be longer for return fixation locations in visual search tasks (see S24 Fig for fixation duration analysis separating target and non-target locations during visual search. (**B**) Saccade size (same format as **A**). Return fixations tended to follow smaller saccades.

Return fixation durations were also longer than to-be-revisited fixation durations, especially during visual search. The difference between return fixation durations and to-be-revisited fixation durations ranged from 23.8 ± 15 ms (Monkeys Free Viewing 1) to 70.7 ± 7 ms (Human Visual Search 2). The duration of return fixations was significantly longer than to-be-revisited fixations for all the visual search experiments and the first two free-viewing experiments. There was no consistent relationship between the duration of to-be-revisited fixations and non-return fixations. Therefore, the increased lingering at return locations could not be simply ascribed to a specific content property of the image (by definition, the image content at to-be-revisited and return locations is very similar).

We further categorized return fixations depending on whether they were at target or non-target locations during visual search. The return fixation durations were longer at target locations compared to non-target locations (S24 Fig, $p = 0.01$ in the Visual Search 1 experiment; and $p < 0.001$ in the Visual Search 2 and 3 experiments). This observation further suggests that return fixations may be associated with the requirements for object recognition during visual search.

Saccade sizes preceding return fixations were generally smaller than those preceding non-return fixations. The difference in saccade sizes ranged from 0.7 ± 0.2 dva (Monkey Free Viewing 1) to 4.3 ± 0.6 dva (Human Videos 2) (Fig 4B and S6(A) Fig). The visual search on object arrays experiment did not show this effect, perhaps because subjects tended to fixate on the objects, which were isolated with no background, and thus there was only a limited repertoire of possible saccade sizes given the geometry of the display. There was also no difference in saccade sizes in the Monkey Free Viewing 2 experiment.

Another important aspect to assess return fixations was the turning angle across three consecutive fixations. We computed two distributions of turning angles based on whether the third fixation was a return fixation or a non-return fixation (Fig 5 and S7 Fig). If subjects looked at location A, continued to look at B and then immediately returned to look at A (a return offset of 1), the turning angle would be 180 degrees. Since we observed that subjects promptly revisit return fixation locations with an offset of 1, we would expect the distribution of turning angles to peak around 180 degrees. Indeed, there was a strong peak at 180 degrees in all experiments for return fixations (Fig 5A and S7(A) Fig) The turning angles for non-return fixations showed a slight U-curve shape with slightly more prominent turning angles of 0 degrees (continuing along the same direction), and 180 degrees (reversing directions) (Fig 5B and S7(B) Fig). The prominence of 0 degrees turning angles may reflect inertia of eye movements [39]. Additionally, saccade sizes are positively correlated with the turning angles [16] (also see S23 Fig). Subjects tend to make shorter saccades when they move their eyes forward.

In sum, return fixations were distinct from non-return fixations and also distinct from to-be-revisited fixations. Subjects tended to remember previously visited locations, reverse eye movement direction, and revisited locations shortly after their first encounter, typically after making a shorter saccade, and generally spending an additional $\sim 50$ milliseconds the second time around.

## Subjects returned more often to salient locations and to locations more similar to the target during visual search

Beyond the proximity to recently explored locations, we asked whether features in the image had an impact on the locations of return fixations. The consistency between subjects described in S4 Fig and S5 Fig suggested that there were special locations in the image that tended to be revisited more often. In addition, the distribution of all return fixation locations was not uniform, further suggesting that there are spatial biases that impact the return fixation locations

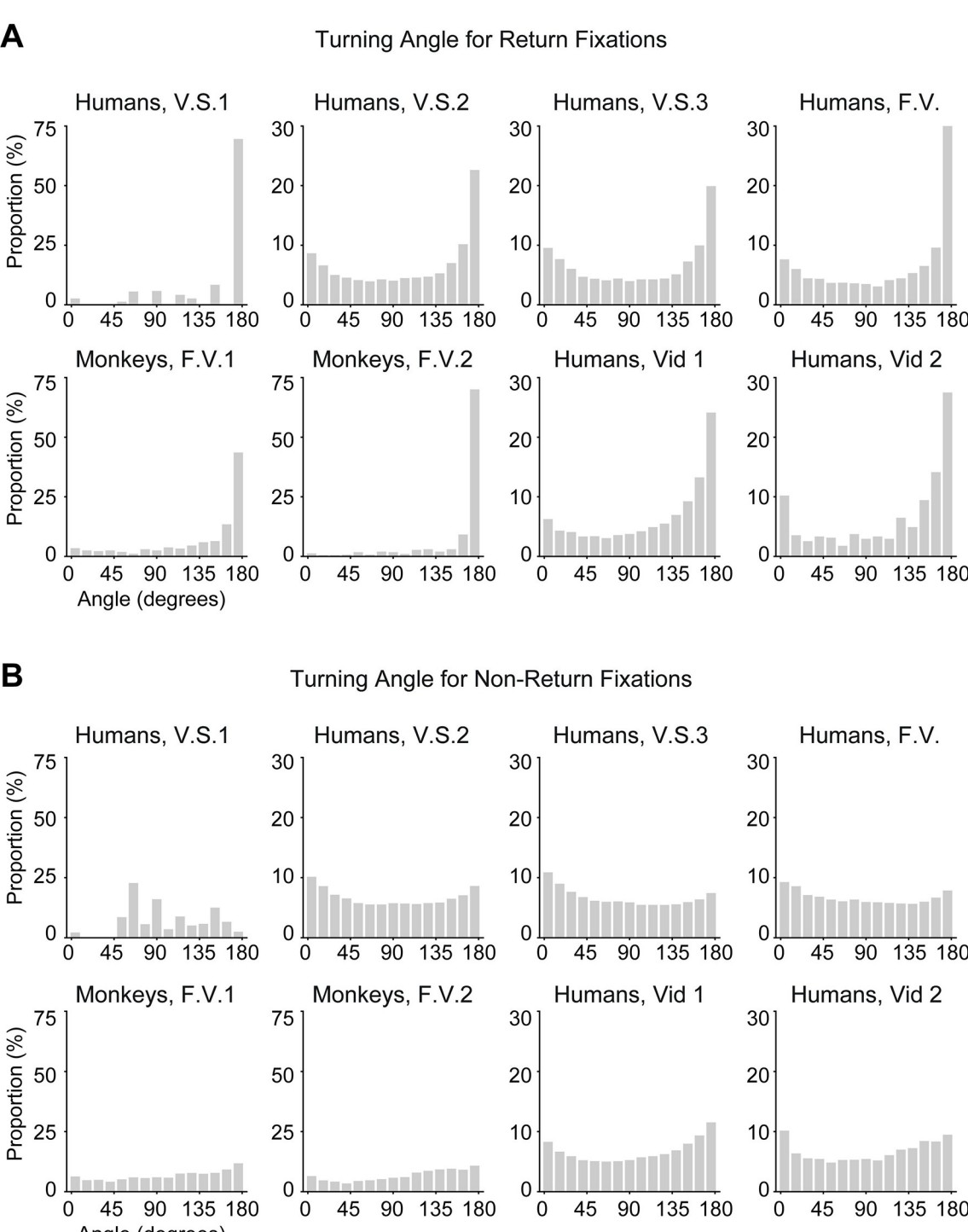

**Fig 5. Subjects tend to make more saccades with turning angles of 180 degrees preceding return fixations.** (**A**) Distribution of turning angles preceding return fixations. Bin size = 12 degrees. The sharp peak at 180 degrees shows that subjects often revert directions to return to their previously fixated location with short offsets (Fig 3). (**B**) Distribution of turning angles preceding non-return fixations. Bin size = 12 degrees. The curves tend to show a slight U-shape showing that subjects often either continue to move in the same direction or else move in the opposite direction.

(S8 Fig). For example, there was a center bias, especially during free-viewing tasks for both humans and monkeys (S8(D)–S8(F) Fig), and the return fixation locations were skewed towards the bottom part of the image in egocentric videos (S8 Fig (G-H)). To test whether these location biases are specific to return fixations or general to all fixations, we plotted the spatial biases for both return fixations (S8 Fig (A1-H1)) and non-return fixations separately (S8 Fig (A2-H2)). We compared the spatial distribution of return versus non-return fixations using the Kullback–Leibler divergence (KLD, which has a value of 0 for two identical distributions and a large value when the two distributions are very dissimilar). To accommodate the sparsity in the fixation distribution and to account for the resolution of the eye tracker, we quantized the fixation locations with a 2D grid of resolution 1 dva, and computed the KLD between return and non-return fixations. The KLD values show that the return fixation distribution tended to be more spatially biased than non-return fixations.

Previous studies have shown that fixations tend to cluster in locations with high bottom-up saliency, such as regions of high contrast changes [10, 11, 50]. Therefore, we asked whether return fixations were distinct in terms of their bottom-up saliency. We used low-level image features, including edges, contrast, intensity, and color, defined in reference [51] to calculate the bottom-up saliency at each location in each image and compared the bottom-up saliency at return fixation locations versus non-return locations. In all experiments except for Visual Search 1, saliency at return fixation locations was higher than at non-return locations (Fig 6A).

Although saliency was higher at return fixation locations, the difference in saliency seemed to be too small to fully explain the pattern of return fixations, especially during visual search conditions. In particular, in the visual search experiment 1, return fixations could not be distinguished from non-return fixations in terms of their bottom-up saliency. We hypothesized that the decision-making process driving return fixations during visual search might also incorporate top-down information, leading us to investigate how the task demands impacted the locations of return fixations. First, we separately considered the sought target location versus non-target locations. In the three visual search experiments (Fig 2A–2C), there were more frequent return fixations to target locations than to non-target locations (Fig 6B). In the visual search experiments in Fig 2B–2C (but not in the object array experiment in Fig 2A), subjects had to use the computer mouse to click on the target location. Therefore, return fixations to target locations most likely imply that subjects fixated on the target but were unaware that they had found it, moved their eyes to other locations, and then returned to the target location, became aware that they had finished the search, and clicked the target location with the mouse.

Even though returning to the target location makes sense in terms of the task goals, subjects also returned to non-target locations. In all three visual search experiments, the proportion of return fixations was higher than expected by chance both for target and non-target locations (Fig 6B). In particular, we compared the Visual Search 2 and Human Free Viewing tasks, which used the same images. The proportion of non-target return fixations in the Visual Search 2 task was larger than the proportion of return fixations in the Human Free Viewing task ($p < 10^{-4}$, two-tailed t-test, $df = 23$). Subjects revisit more locations during visual search compared to free-viewing conditions, even when those locations do not contain the target.

A simple hypothesis of why subjects may return to non-target locations is that those locations may share some degree of visual similarity with the target, based on the previous visual search work [8]. To test this hypothesis, we designed an experiment to assess the degree of visual similarity between different fixation locations with the sought target (Fig 6C). Subjects were presented with the target image plus two options and were asked to choose the image that was most visually similar to the target (Methods). The subjects participating in these two psychophysics experiments on target similarity were different from the ones in the two original

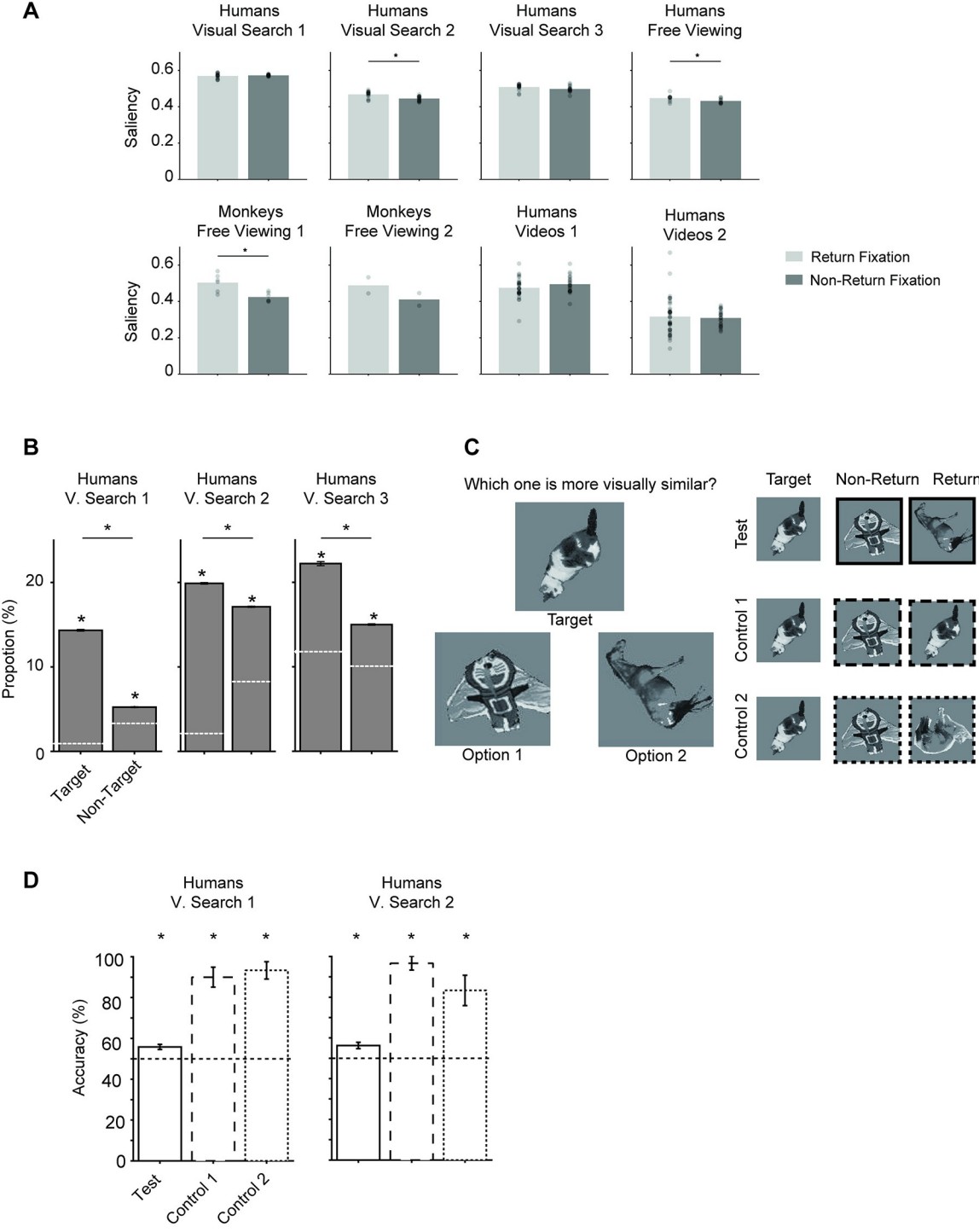

**Fig 6. Return fixation locations depended on the image and task.** (**A**) Return locations tend to show higher saliency. Each dot shows a different subject. * denotes $p < 0.05$, two-tailed t-test. To-be-revisited and return locations overlap by definition, saliency at to-be-revisited locations was similar to that at return locations.(**B**) During visual search tasks, the proportion of return fixations was larger for target than non-target locations. Gray asterisks denote a statistically significant difference in the proportion of return fixations with respect to the chance levels. Black asterisks denote a statistically significant difference between target versus non-target locations. (**C**) Schematic of experiment to assess whether return fixations shared similarity to the target. In each trial, subjects were presented with an image (Target) and had to choose the more similar image between two alternatives. There were three conditions: (1) Test, where the two alternatives were return fixations versus non-return fixations, (2) Control 1, where the two alternatives were an identical copy of the target versus non-return fixations, and (3) Control 2, where the two alternatives were a different object from the same category as the

target versus non-return fixations. (**D**) Accuracy in distinguishing images from each of the conditions in **C** versus non-return fixations. Asterisks denotes statistically significant difference from chance levels (horizontal dashed line at 50%, ($p < 0.05$, one sample t-test)). Photo sources: (C) was modified from a public dataset [8].

visual search tasks. To ensure that subjects performed the task as directed, we included two controls where one of the options was identical to the target (control 1), or one of the options was a different exemplar from the same category (control 2). As expected, subjects chose the control images over non-return fixations (Fig 6D). In the test condition, one of the images was a non-target return fixation and the other one was a non-return fixation. Subjects indicated that the return fixations were slightly more similar to the target than non-return fixations 55.8 ± 1.25% of the time in the visual search experiments 1 ($p = 0.002$, one-sample t-test comparing to chance, $t = 4.45$, $df = 9$) and 56.3 ± 1.57% of the time in the Visual Search 2 experiment ($p = 0.003$, one-sample t-test comparing to chance, $t = 3.99$, $df = 9$). In sum, subjects returned more frequently to salient locations, to locations containing the target, and to locations resembling the target in visual search experiments.

## A generalist computational model of eye movements revealed return fixations

To further understand the mechanisms that give rise to return fixations, we developed an image-computable model capturing the basic observations in Figs 3–6. A schematic diagram of the model is shown in Fig 7. The starting point was the neurophysiologically-inspired invariant visual search network (IVSN) [8]. IVSN consists of a "ventral visual cortex" module, implemented by a pre-trained deep convolutional neural network (the VGG-16 network, [52]), and a "prefrontal cortex" module. The visual features from the target image are temporarily stored in prefrontal cortex and modulate the features of the search image in a top-down fashion, creating a target feature similarity map, $M_{sim}$ (S9(A) Fig and S11(D) Fig). The IVSN model uses this map to generate a sequence of fixations. The model does not have any mechanism to process motion information or integrate temporal information across video frames; therefore, we focus here on modeling the results of the first six experiments on static images (Methods).

Several modifications were introduced into the IVSN architecture. First, to produce a sequence of eye movements during free-viewing conditions, we incorporated the possibility of having uniform top-down modulation [8] and introduced a bottom-up saliency map [53, 54], $M_{sal}$ (S9(A) Fig and S11(C) Fig, Methods). The saliency map ($M_{sal}$) depended exclusively on the image contents, while the similarity map ($M_{sim}$) additionally depended on the target during visual search. Of note, the weight parameters used for extracting visual features for $M_{sim}$ and $M_{sal}$ were neither trained with any of the images used in this study, nor were they trained to match human performance: all the weights in the VGG-16 architecture were pre-trained using the ImageNet dataset in a visual recognition task [52].

Second, we incorporated a constraint on saccade size [55, 56]. The distribution of saccade sizes in humans and monkeys is not uniform (S6(A) Fig): eccentricity-dependent sampling and oculomotor constraints imply that there are few large saccades and the saccade sizes follow an approximately gamma distribution [57]. Therefore, for each fixation $t$, we included a saccade prior map, $M_{sac,t}$, computed from the current fixation location and an empirical saccade size distribution for each task (S10(B) and S11(E)) Figs. Thus, in contrast to the previous two maps, $M_{sac,t}$ does not depend on the image content.

A third and critical modification is the introduction of a memory decay function for previous fixation locations [17, 58, 59]. Many visual search models, including the initial

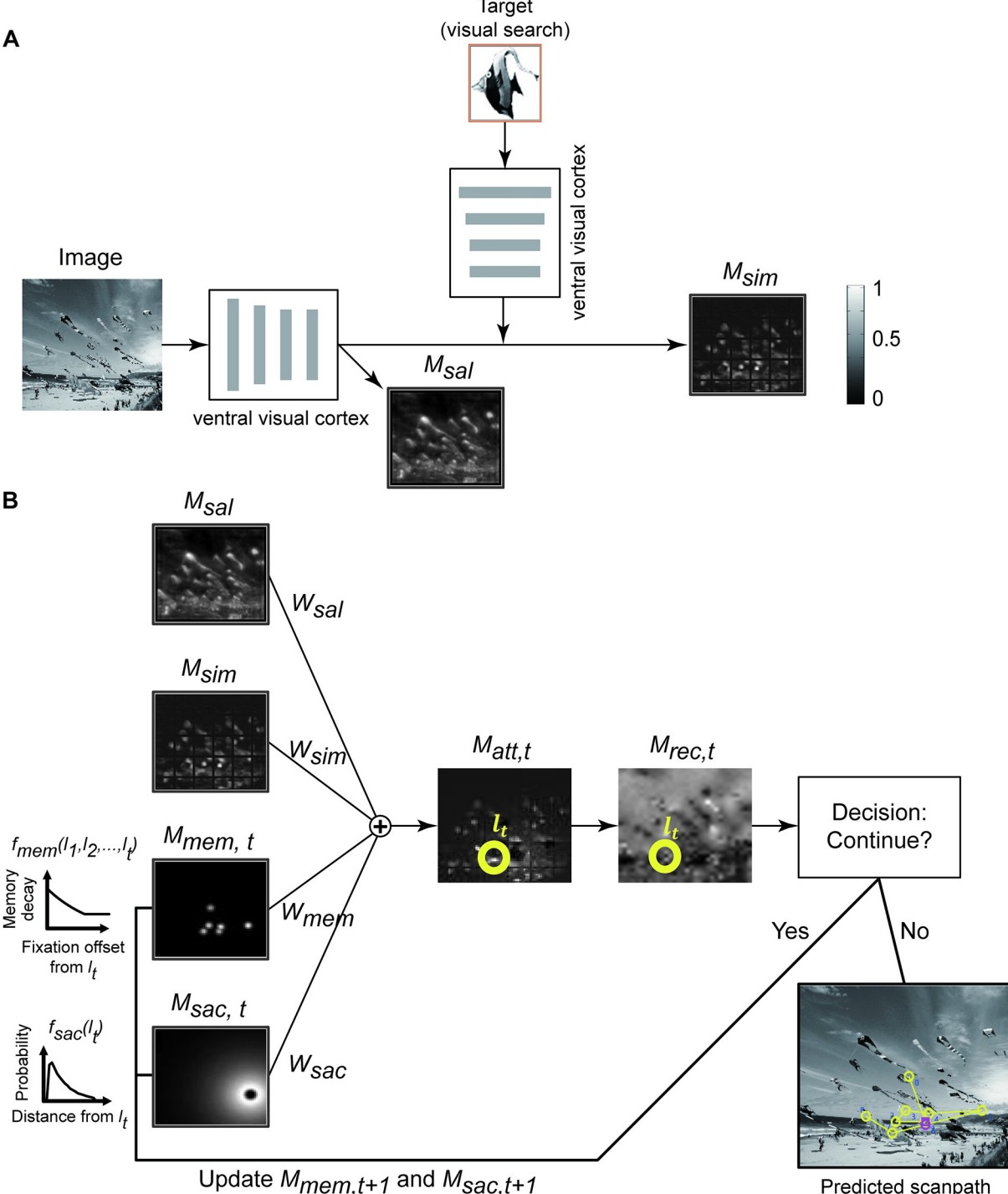

**Fig 7. Architecture of the computational model.** (**A**) The model has a ventral visual cortex module (pre-trained VGG-16) that extracts features from the image. These features constitute the saliency map ($M_{sal}$). In visual search tasks, the same ventral visual cortex module processes the target image and modulates the features in the search image via top-down modulation, generating a target similarity map ($M_{sim}$) [8]. See S9 Fig and Methods for detailed architecture. (**B**) The generation of eye movements is governed by a weighted combination of 4 maps: $M_{sal}$ (part **A**), $M_{sim}$ (part **A**, only in visual search tasks), a time-dependent memory map ($M_{mem,t}$, and a time-dependent saccade size constrain map ($M_{sac,t}$). At each time point $t$, the 2D spatial memory decay map $M_{mem,t}$ is updated based on all previous visited locations $\{l_1, l_2, \ldots, l_t\}$. The brighter the color on the memory decay map, the stronger the effect of memory inhibition. The saccade size constrain map is also updated at each time point according to the current fixation location, $l_t$. A winner-take-all chooses the maximum in the combined attention map $M_{att,t}$ (yellow circle) as the location for the next fixation. In visual search tasks, the model computes a recognition map $M_{recog}$ indicating the confidence that the current

fixation contains the sought target (S9(B) Fig). If the target is found, the search stops; otherwise, it continues. During free viewing tasks, the recognition map is not used and the model keeps generating eye movements for a fixed amount of time. Photo sources: (A, B) were modified from a public dataset [8].

implementation of IVSN include an infinite inhibition-of-return mechanism that prohibits fixations to revisit previous locations. Instead, we introduced a memory decay map $M_{mem,t}$ that contained information about previously visited locations (S10(A) and S11(F)) Figs. The map $M_{mem,t}$ does not depend on the image contents but rather it is calculated from the eye positions at all the previous fixations $1, \ldots, t$.

The model linearly combines the four maps, $M_{sim}$, $M_{sal}$, $M_{sac,t}$ and $M_{mem,t}$, producing a final attention map, $M_{f,t}$. (Methods, S11(G) Fig). The linear combination involved two scalar weights, $w_{sac}$ and $w_{mem}$, which control the relative importance of the $M_{sac,t}$ and $M_{mem,t}$ maps. We set values for $w_{sac}$ and $w_{mem}$ based on the Visual Search 2 experiment. These two parameters were then fixed and used to test all the datasets. The location of the next fixation for the model is dictated by the maximum of $M_{f,t}$ (S11(G) Fig). The saccade and memory maps are updated after each fixation (the similarity and saliency maps are fixed), and a new fixation is generated. During visual search, the model decides whether the current fixation contains the target or not by using the ventral visual cortex to extract visual features in the current fixation and comparing those features to the stored target features (S9(B) Fig, Methods). The process is iterated until the target is found in visual search tasks or for a fixed number of steps in the free viewing tasks.

An example sequence of fixations produced by the model in the free-viewing experiment is shown in Fig 8A, right. The model makes one return fixation denoted by the red triangle, which is close to the location of a return fixation made by the monkey in the same experiment. In this example, both the monkey and the model revisited a location within the face (note that the model has no special bias towards faces and follows features extracted from the image by the ventral visual cortex module). Further visualization examples for each of the six datasets are shown in S12 Fig.

Similar to the results shown for humans and monkeys in Fig 3, the model made more return fixations than expected by chance in all six experiments (Fig 8B, $p < 10^{-3}$, bootstrap 1000 resamples), with a proportion of return fixations ranging from 5.5 ± 0.4% in the human Free-Viewing experiment to 25.3 ± 1% in the human Visual Search 2 experiment. The proportion of return fixations was higher in the visual search tasks (Fig 8, columns 1–3) compared to the free-viewing tasks (Fig 8, columns 4–6, $p < 10^{-3}$, bootstrap 1000 resamples).

Consistent with the results shown in Fig 6B, even though the model does not explicitly incorporate any target location information, the model tended to produce a higher proportion of return fixations at the target locations than at non-target locations in the Visual Search 2 and 3 experiments (S13 Fig), but not in the Visual Search 1 experiment.

## The computational model captures key properties of return fixations

Next, we compared the properties of return fixations between humans/monkeys and the model. Consistent with the results in humans and monkeys (Fig 3J), most of the return offsets for the model tended to be small and the return offset distribution showed an approximately exponential decay (Fig 8C).

The model does not have any notion of fixation duration and therefore we cannot plot the equivalent to Fig 4A. The overall distribution of saccade sizes for the model was similar to the one for humans and monkeys (compare S6(A) Fig versus S6(B) Fig). In the Visual Search 2 and 3 tasks, the saccade sizes preceding a return fixation for the model tended to be larger than

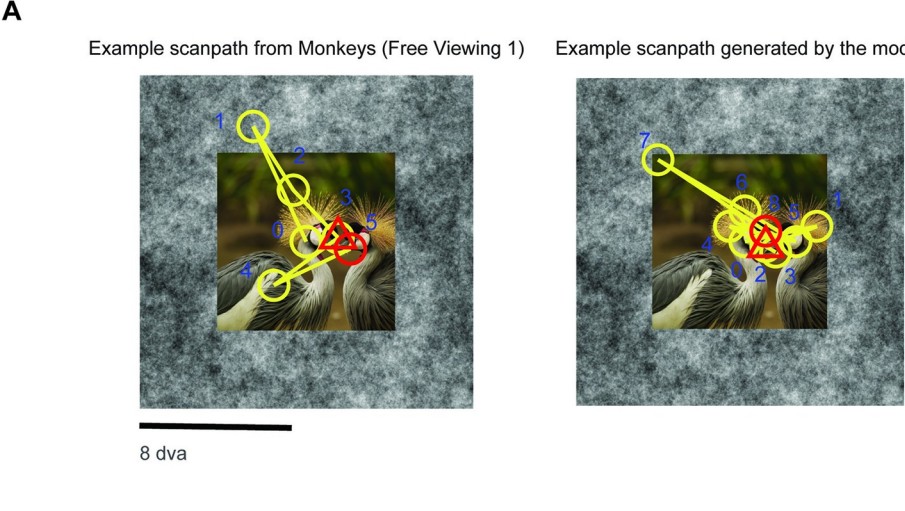

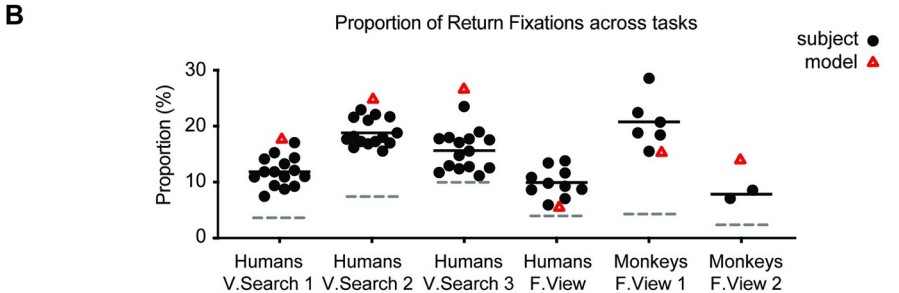

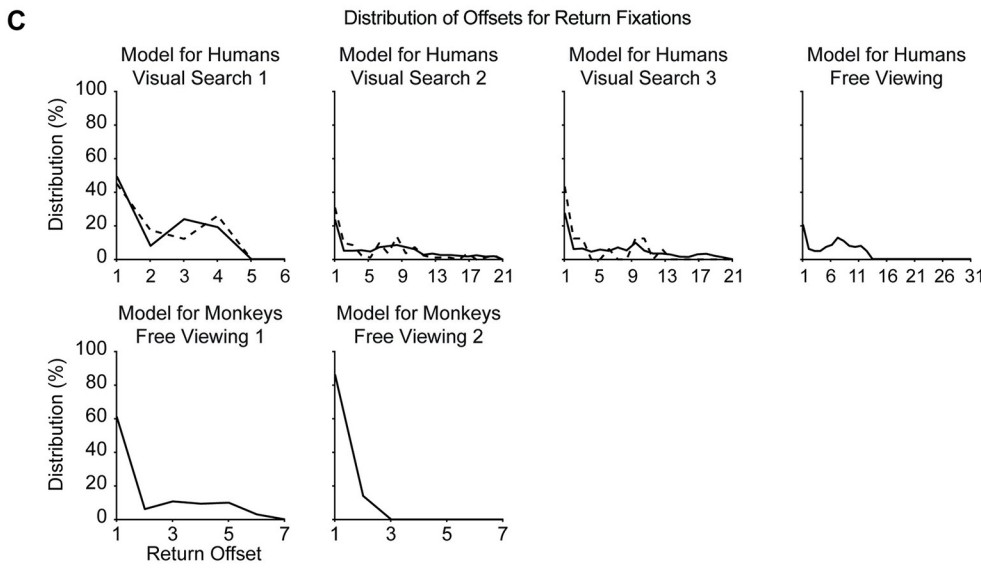

**Fig 8. The computational model generates return fixations.** (**A**). Example scanpaths by monkeys and by the model while free viewing natural images (conventions follow those in Fig 1.) (**B**) Proportion of return fixations. The black dot denotes the proportion of return fixations for each subject (reproduced from Fig 3A–3H). The red triangle denotes the proportion of return fixations for the model. Black solid lines show average across subjects and gray dashed lines show chance level. (C) Distribution of the return offset for the model (format as in Fig 3J). Photo sources: (A) Reproduced with permission from BPRC and pexels.com.

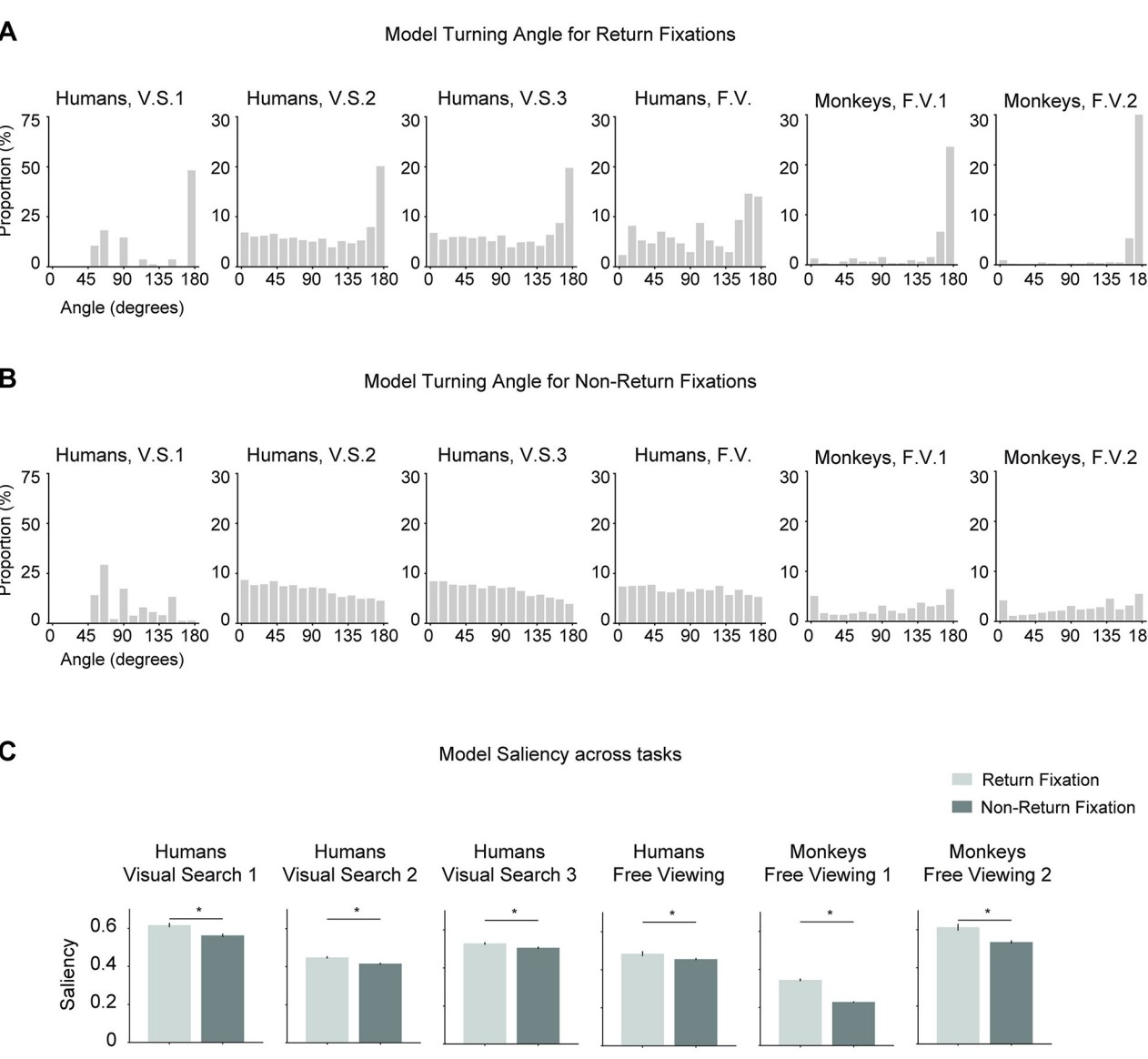

**Fig 9. The computational model captures return fixation turning angles and saliency.** Distribution of turning angles preceding return fixations (**A**) and non-return fixations (**B**) for the model (format as in Fig 5A–5B). (**C**). Saliency at return fixation locations and non-return fixation locations for the model (format as in Fig 6A).

those preceding non-return fixations (S14 Fig, columns 1–3), which is different from the trend observed in humans in Fig 4B. In the human and monkey free-viewing tasks, the saccade sizes preceding return fixations for the model tended to be smaller (S14 Fig, columns 4–6), consistent with the results in humans and monkeys Fig 4B.

Image properties also influenced the probability of making a return fixation to a particular location in the model, as shown by the increased bottom-up saliency at return fixations compared to non-return fixations (Fig 9C, compare to Fig 6A). The model also captured the sharp peak of 180-degree turning angles (compare Figs 5A and 9A; see also S15(A) Fig). In the case of non-return fixations, the model is also qualitatively similar to human and monkey behavior (compare Figs 5B and 9B; see also S15(B) Fig).

It is important to emphasize that there were no free parameters in the model tuned to reproduce the properties of return fixations in the experimental data. In sum, without any task-specific training, the model approximated most of the basic properties of return fixations in humans and monkeys.

## Ablations highlight the relevance of each model component to capture return fixation properties

To gain further insights about the model design choices and their impact on return fixations, we conducted ablation studies, separately removing either the similarity/saliency module (**Ablated similarity/saliency**), the memory module (**Ablated memory**), or the saccade distribution module (**Ablated Saccade Distribution**). Details about the implementation of the ablated models are described in the Methods section. For each ablated model, we recomputed the proportion of return fixations (S16 Fig), the return offsets (S17 Fig), the saccade size distribution (S18 Fig), the turning angles (S19 Fig), and the saliency at fixated locations (S20 Fig). Table 1 summarizes the key ablation results.

As expected, ablating the memory module led to large increase in the proportion of return fixations (Table 1, S16 Fig). Conversely, and trivially, a model with infinite inhibition of return demonstrates no return fixations [8].

Removing the similarity/saliency module led to more random fixations and to a large reduction in the proportion of return fixations (Table 1, S16 Fig). During the visual search experiments, saccade locations are largely dictated by the similarity to the target rather than bottom-up saliency. Conversely, during free viewing, saccade locations are largely dictated by saliency. The distribution of turning angles for return fixatons was also severely distorted in the absence of the similarity/saliency module (Table 1, S19 Fig). As expected, the image patch saliency at fixated locations was reduced when this module was ablated (Table 1, S20 Fig).

The saccade size constraint had a lesser impact on the overall proportion of return fixations (S16 Fig), but removing this constraint led to a much more uniform distribution of saccade sizes, which is inconsistent with the experimental data (Table 1, S18 Fig). Removing this constraint also led to longer return offsets than observed experimentally (Table 1, S17 Fig).

In sum, altering each of the modules led to worse correspondence to the experimental properties of return fixations. These ablation studies suggest that return fixations in humans and monkeys are orchestrated by multiple interacting mechanisms.

**Table 1. Summary of similarity indices for full and ablated versions of the model for multiple properties.** Values indicate the mean similarity index (with SD in parentheses) across the six tasks (Human Visual Search 1, 2, and 3, Human Free Viewing, and Monkey Free Viewing 1, and 2). Highlighted in bold are the values that showed large differences with respect to the full model. These results indicate that altering each of the model modules led to worse correspondence to the experimental properties of return fixations. For each property, the supplementary figures cited in the first column provide similarity indices for each individual task.

| | Full Model | Abla. Sacc. Dist. | Abla. Memory | Abla. Sim/Sal. |
|---|---|---|---|---|
| Prop. Ret. Fixations (S16(B) Fig) | 0.74 (0.08) | 0.75 (0.13) | **0.53 (0.13)** | **0.19 (0.13)** |
| Return Offsets (S17(B) Fig) | 0.63 (0.11) | **0.53 (0.07)** | 0.71 (0.07) | **0.48 (0.29)** |
| Saccade Dist. (S18(B) Fig) | 0.69 (0.16) | **0.49 (0.12)** | 0.71 (0.17) | **0.49 (0.26)** |
| Turn. Angle, Ret. Fix. (S19(C) Fig) | 0.73 (0.15) | 0.72 (0.12) | 0.75 (0.12) | **0.41 (0.32)** |
| Turn. Angle, NonRet. Fix. (S19(C) Fig) | 0.87 (0.07) | **0.73 (0.10)** | 0.87 (0.08) | 0.84 (0.11) |
| Saliency, Ret. Fix. (S20(B) Fig) | 0.93 (0.07) | 0.93 (0.08) | 0.92 (0.12) | **0.81 (0.11)** |
| Saliency, NonRet. Fix. (S20(B) Fig) | 0.91 (0.12) | 0.91 (0.13) | 0.91 (0.11) | **0.82 (0.16)** |

## Discussion

We examined and modeled 44,328 return fixations out of a total of 217,440 fixations (20.4%). In contrast to previous studies focusing on a single task and specific conditions in humans, here we studied eight experiments monitoring eye movements across a wide range of tasks and experimental conditions in humans and monkeys (Fig 2). Return fixations were ubiquitous across visual search tasks of different complexity levels, during free-viewing conditions, in humans and monkeys, and also during naturalistic freely moving behaviors (Fig 3). Return fixations tended to occur shortly after the first visit to a given location, in many cases with the minimum possible offset of one intervening fixation (Fig 3J). Return fixations lasted ∼50 ms longer (Fig 4A), often following small saccades (Fig 4B) and a shift in the saccade direction (Fig 5). The locations of return fixations were consistent across subjects (S4 Fig and S5 Fig) and tended to cluster around regions of higher bottom-up saliency (Fig 6A) as well as towards regions that resemble the target during visual search tasks (Fig 6B–6D).

We developed an image-computable neural network model that simulated the universal properties of return fixations (Figs 7 and 8 and 9) in multiple tasks from multiple species. The proposed model has five key components: (i) an image feature extractor, (ii) a saliency map, (iii) a target similarity map, (iv) a constraint on saccade sizes, and (v) a finite inhibition-of-return with an approximately exponential memory decay function (Fig 7B). The first two components depend exclusively on the image contents. Salient locations include spatial changes in color, orientation, and texture, among other properties. There is extensive literature documenting the role of salient locations in attracting eye movements [53, 54]. The target similarity map, which is only relevant during visual search tasks, makes image locations that resemble the target especially attractive for fixations [8] (S16 Fig and S20 Fig). We used convolution neural networks to extract image features. These neural networks preserve spatial locations when extracting visual features, and that helps us easily align and combine the different attention maps, and ultimately apply the Winner-Take-All mechanism. In the brain, there could be multiple maps in different neural circuits and with different resolutions [60], raising the question of how these different maps are combined.

The third model component is based on the observed distribution of saccade sizes (S6 Fig), which favors small saccades. The avoidance of large saccades is likely to be due to a combination of eccentricity-dependent sampling and constraints imposed by the eye movement musculature itself [55–57, 61]. This constraint also makes it more likely to revisit recent locations (Fig 4B), but previous work has shown that the saccade size distribution is not sufficient to account for the frequency of return fixations [39] (S17 Fig and S18 Fig).

The fourth model component is memory decay, which approximates finite inhibition-of-return (IOR). The strength of IOR plays a central role in balancing curiosity towards new locations (stronger inhibition of prior fixations enhances foraging of novel image locations) versus scrutiny of known locations (weaker inhibition facilitates return fixations). The proportion of return fixations is lower than what would be expected by a memoryless system [39] (S16 Fig), which can be explained by a finite IOR. Although a finite IOR has been partially attributed to memory capacity limitations, it can also bring benefits in biological vision. Because recognition during rapid saccades in complex and cluttered environments can be imperfect, it may be advantageous to return to previous locations. Furthermore, it may also be useful to frequently inspect important or high-risk areas, such as regions where predators are likely to hide. Similarly, in real-world, dynamic environments, the environment is continually updated as items move, the subject navigates and takes action, and regions become occluded or revealed. Finite IOR allows the subject to review previously-visited locations that may have dynamically changed. As an initial approximation, the computational model assumes that the finite IOR

function is fixed and independent of the task, species, or experimental conditions. Differences in the return fixation properties across tasks are thus accounted for in the model by the integration of the four components.

Task demands can play an important role in determining the frequency and duration of return fixations. For example, during the cooking task when the subject is cutting carrots (Fig 2G), the eyes are constantly drawn to the knife and carrot, thus increasing the number of return fixations. Under these conditions, there is no need to process extensive information at each fixation and the fixation durations are shorter. In contrast, during the Waldo search task (Fig 2C), there is a stronger incentive for exploring novel locations, thus reducing the frequency of return fixations, yet each return fixation lasts longer as the large amount of clutter makes the target recognition decision harder. Task instructions are also likely to alter the frequency and properties of return fixations. For example, if subjects are instructed to fixate each location for a long time (on the order of seconds), there may be less incentive to return to this location and the return offset might be longer.

It is interesting to speculate that return fixations may be especially linked to an imperfect visual recognition machinery. In an extreme case where the visual recognition machinery achieves perfect performance in interpreting the contents at the fovea in every fixation, there would be little incentive to revisit locations to gain further insights. Multiple studies have praised the virtues of fast recognition in approximately 150 ms after flashing a stimulus [62–64]. However, many of those studies have focused either on isolated objects or large objects with minimal clutter. Under more natural conditions and especially for smaller objects embedded in clutter, subjects make many recognition mistakes [65, 66]. Indeed, a strong example of recognition errors is the case of return fixations to the target during visual search (Fig 6B; see also discussion in [8]). Consistent with the link between return fixations and imperfect recognition during a single fixation, several studies have argued that return fixations allow reinspection of incomplete or dynamic regions in scenes [39], recovery of lost information [35–38], and rehearsals of visual working memory [31, 34].

The proposed model is deliberately founded on using pre-trained neural networks and has very few free parameters. The weights to extract image features in the ventral visual cortex module are pre-trained on an independent visual recognition task using the ImageNet dataset and are *not* fine-tuned for any of the images or tasks in the current study. The saccade size constrain, the IOR memory decay function and the relative weight of those two components are derived from experimental data. Specifically, they were tuned using the Visual Search 2 experiment. All of those parameters were fixed thereafter, and the results for all the other datasets shown throughout the text do not use any type of data fitting. The model does not always quantitatively match the observations in humans and monkeys (e.g., compare Fig 3B versus 8B, Fig 4B versus S14 Fig, Fig 6A versus 9C). Fine-tuning the model for each dataset would improve the quantitative fitting to each experiment, allow to incorporate task-specific instructions, and also to capture differences between individual subjects. However, the purpose of the model was to provide a conceptual proof-of-principle demonstration of the key ingredients underlying return fixations rather than an attempt at fine-tuning multiple parameters to fit the experiments. Thus, without any explicit training or data fitting, the computational model makes testable predictions about viewing behavior across many complex tasks and species.

Most eye movement studies have focused on flashing two-dimensional images on a computer screen. This paradigm as a surrogate for natural vision has been criticized for lacking depth information, natural spatiotemporal statistics, and natural head and body movements. Egocentric videos provide a more naturalistic venue to study first-person viewing behaviors where subjects interact with physical objects while freely moving their eyes, heads and bodies. Despite the notable experimental differences to flashing images, in terms of return fixations,

subjects revisit previously fixated locations during naturalistic behaviors like visual search and cooking, in a similar fashion to the behavior observed in static images. Task demands and natural behaviors may impose additional constraints. Several studies have shown that the egocentric gaze in a natural environment requires the combination of gaze direction (the line of sight in a head-centered coordinate system), head orientation, and body pose [67–69]. For example, during the cooking task, the gaze point tends to fall on the object that is currently being manipulated. This constraint results in a higher proportion of return fixations compared with the other tasks.

The amount of time devoted to visual processing during a saccade has been used as a proxy for the computational demands of given tasks [70, 71]. Consistent with this notion, the average fixation duration during free viewing (259.7 ± 0.9 ms) was shorter than during visual search (297.9 ± 0.9 ms). Interestingly, return fixations showed longer durations (Fig 4A). A possible interpretation of this observation is that the brain tags these locations as return fixations, perhaps acknowledging the difficulties in visual recognition during the first pass, and devotes additional computational time to improving recognition the second time around.

Even though the model captures essential properties of return fixations, the model does not behave exactly the way humans and monkeys do. First, the model shows constant acuity over the entire visual field, which is clearly not the case for primate vision where acuity drops rapidly from the fovea to the periphery [72]. Second, humans and monkeys have a better recognition system to decide whether the target is present or not at the current fixation (S21 Fig). Third, humans and monkeys may capitalize on contextual information integrated over multiple saccades to decide whether a return fixation is warranted or not [66]. For example, the object relations in the environment might attract primates to check back for relevant objects. Fourth, there is no learning in the current model, but humans and monkeys can adapt and change their strategies in a task-dependent manner. Fifth, the model does not capture the duration of each fixation or saccade; nor does the model account for the momentum carried from a previous saccade, which also plays an important role in constraining primate eye movements.

Despite these limitations, the model provides a simple, reasonably good, and plausible mechanistic account of the key components responsible for return fixations: bottom-up saliency, top-down task goals such as target similarity, a constrain on saccade sizes, and a finite memory. The model shows return fixations and reproduces the fundamental properties of return fixations, including their frequency, offsets, turning angles, and preferred image locations. The model can capture these properties across different tasks, different species, and different experimental dynamics and conditions, without any retraining or parameter fitting in each condition. Given the ubiquitous presence of return fixations, these computational efforts open the doors to help build better models of active scene sampling via eye movements during visual search and visual object recognition.

## Methods

### Ethics statement

All the human psychophysics experiments were conducted with the subjects' written informed consent and according to the protocols approved by the Institutional Review Board at Boston Children's Hospital. All animal research procedures were approved by the Harvard Medical School Institutional Animal Care and Use Committee, and the Washington University School of Medicine Institutional Animal Care and Use Committee, and conformed to NIH guidelines provided in the Guide for the Care and Use of Laboratory Animals.

## Datasets

We evaluated return fixations on eight datasets (Fig 2), individually described below.

**Visual search on static images.** We evaluated eye movements during three visual search tasks with increasing level of difficulty, reported in reference [8]: object arrays (Visual Search 1, Fig 2A), natural images (Visual Search 2, Fig 2B), and Waldo images (Visual Search 3, Fig 2C). Forty-five naive observers (19–37 years old, 15 subjects per experiment) participated in these tasks. Subjects had to fixate on a cross shown in the middle of the screen for 500 ms, a target object was presented followed by another fixation delay (object arrays and natural images), a search image was presented, and subjects had to move their eyes to find the target. In the natural images (Visual Search 2) and Waldo images (Visual Search 3), subjects had to indicate the target location via a mouse click. If the clicked location fell within the target, subjects went on to the next trial; otherwise, subjects stayed on the same search image until the target was found. If the subjects could not find the target within 20 seconds, the trial was aborted and the next trial was presented. For further details about the images, the eyetracking experiment setup, and the tasks, see reference [8].

In these visual search experiments and also in the free-viewing experiments described next, the participants' eye movements were recorded using the EyeLink 1000 plus system, sampling at 500 Hz and <1 dva resolution (SR Research, Canada). All participants had normal or corrected-to-normal vision. Participants were compensated for participation in the experiments. All the human psychophysics experiments were conducted with the subjects' written informed consent and according to the protocols approved by the Institutional Review Board at Boston Children's Hospital. Stimuli were presented in grayscale on a 19-inch CRT monitor (Sony Multiscan G520) occupying full screen (1024 × 1280 pixels, subtending 25 × 30 degrees of visual angle (dva)). Observers were seated at a viewing distance of 66.4 cm.

**Human free viewing.** To compare human eye movements during visual search versus free viewing, we conducted an experiment with 10 subjects (18–37 years old, 5 female). We used the same 240 natural images from the Visual Search 2 task, described above. Subjects had to first fixate on the center cross for 500 ms and then freely move their eyes to explore the image for 4500 ms (Fig 2D). The stimulus presentation duration of 4500 ms was chosen because subjects were able to find the target in 90% of the trials within this time during the visual search task (Visual Search 2) [8], therefore providing us with an approximately comparable number of fixations. Subjects were instructed to look at the images, without any other task demands.

**Monkey free viewing.** To compare eye movements under free viewing conditions across species, we analyzed eye-tracking data from free-viewing monkeys (Fig 2E–2F). These two data sets were initially collected for other experiments with the goal of studying neuronal responses during free viewing; the neuronal responses are not discussed here. These two experiments were not designed to specifically match the conditions in the previous experiments. The purpose of introducing these experiments in the current study is not to quantitatively assess whether humans make more or less return fixations than monkeys, which would require matching the experimental conditions. Rather, the purpose here is to qualitatively assess the properties of return fixations in different species and to evaluate whether the model can capture those properties. Six monkeys (5–13 years old, all male) from one lab were tested in the Monkey Free Viewing 1 experiment. Two monkeys (both 7 years old males) from a second lab were tested in the monkey Free Viewing 2 experiment. Procedures were approved by the Harvard Medical School Institutional Animal Care and Use Committee (Monkey Free Viewing 1), and the Washington University School of Medicine Institutional Animal Care and

Use Committee (Monkey Free Viewing 2), and conformed to NIH guidelines provided in the Guide for the Care and Use of Laboratory Animals.

In the Monkey Free Viewing 1 experiment (Fig 2E), each trial of the experiment did not start with a center cross. Instead, the initial fixation of a trial could start anywhere on the screen. The presentation duration of each trial vary from 1,000 ms to 2,000 ms. There were 121 images in total with repeated presentations in random order. In our analysis, we focused on fixation sequences longer than 1,500 ms during *only* first stimulus presentations. We discarded other fixation sequences in repeated presentations due to concerns about the impact of memory across trials on return fixations. The trial sequence intermixed both natural images and images containing salient visual features, such as other monkeys or body parts. There were 36 images out of 121 containing faces. To ameliorate the center fixation bias in each trial, the image location was randomly jittered relative to the original image in a [-3:1:3] dva grid. For example, in Fig 2E, the stimulus was shifted to the left; with the vacant space shown as grey background. Monkeys were seated at a viewing distance of approximately 57–58 cm away. All images were presented in color on a monitor screen ($635 \times 635$ pixels, subtending $16 \times 16$ dva).

In the Monkey Free Viewing 2 experiment (Fig 2F), monkeys were trained to first look at the center fixation for 500 ms, followed by the stimulus presentation for 1000—1500 ms. There were 1,761 images in total with 1,380 natural images from the MSCOCO dataset [73], around 240 images that contain either monkey faces or their body parts, and around 140 pictures of local laboratory staff and animal shelters. As in the Monkey Free Viewing 1 experiment, to eliminate the center bias, all the images were shifted randomly in a 2-degree radius circle. Monkeys were seated at a viewing distance of approximately 57–58 cm away. All images are presented in color on a monitor screen ($596 \times 596$ pixels, subtending $15 \times 15$ dva).

**Human egocentric videos.** While the majority of eye movement studies have focused on static images, here we also considered a more naturalistic setting where subjects could move freely and interact with physical objects while eye positions and first person videos were recorded (Egocentric videos, Fig 2G–2H). We used two existing egocentric video datasets [48, 49]. In both cases, subjects wore an SMI mobile eyetracker (iMotions, Denmark) with a sampling rate of 30 Hz and precision of $\approx 0.5$ dva.

The egocentric video dataset 1 (Humans, Videos 1) consisted of eye positions in 86 videos showing the field of view captured from a fist-person perspective [48]. In these videos, 32 subjects performed cooking activities. Each video clip lasted 20 minutes and was captured at 24 frames per second and $960 \times 1280$ pixels (corresponding to $\approx 46 \times 60$ dva). In the beginning of each cooking task, subjects were instructed to follow the steps on a recipe. There were 10 recipes, such as northern American breakfast, pizza, and turkey sandwich. Each recipe entailed a sequence of meal preparation steps. S2(B) Fig shows example video frames and their corresponding fixations overlaid on the last frame of each video clip (yellow circles).

The egocentric video dataset 2 (Humans, Videos 2) consisted of eye positions in 57 videos showing the field of view captured from a first-person perspective [49]. In these videos, 44 subjects performed a visual search task. Each video clip lasted around 15 minutes and was captured at 24 frames per second and a resolution of $960 \times 1280$ pixels (corresponding to $\approx 46 \times 60$ dva). The experiment site was a fully furnished and functional model home including a master bedroom, children's room, living room, open kitchen, dining area, study room, recreational room, bathroom and exercise area. Each subject was asked to search for a list of 22 items commonly used in daily life (including thumb drive, shampoo, etc.) and move them to the designated packing location (dining table). S2(C) Fig shows example video frames and their corresponding fixations overlaid on the last frame of each video clip (yellow circles).

## Psychophysics experiments on target feature similarity for return fixations

In Fig 6C and 6D, we asked whether the return fixation locations were visually similar to the sought target during the visual search tasks. To answer this question, we conducted two psychophysics experiments on Amazon Mechanical Turk (Mturk). We recruited 20 subjects (10 subjects for the object arrays task and 10 subjects for the natural images task).

Subjects were presented with a target object and two alternative images. Subjects performed a two-alternative forced choice task indicating which of the two options was more similar to the target object. The images remained on the screen until the subjects made a choice. The two image options were randomly mapped onto choice A or B. The images were fixation patches obtained by cropping the search image. In the main test condition, one of the options always corresponded to a return fixation patch and the other option corresponded to a non-return fixation patch, where return and non-return were defined based on the eye movement data independently obtained from different subjects in reference [8]. For each trial, the return and non-return fixation image patches were extracted from the same image and trial. In the object array experiment (Fig 2A), the patch encompassed the entire object (subtending about 3.6 dva). In the natural images experiment (Fig 2B), the patch encompassed a square box of size $156 \times 156$ pixels, subtending 3.6 dva and centered at each fixation. We collected 412 pairs on object arrays and 1,041 pairs on natural images.

As a sanity check to evaluate the quality of the online results from Mturk, we introduced two control conditions that were randomly intermixed with the test trials. In control 1 (3% of the trials), the two options were a non-return fixation patch versus the actual identical target. We (obviously) expected subjects to indicate that the identical target was more similar to the target than the non-return fixation locations. In control 2 (3% of the trials), the two options were a non-return fixation patch versus an object belonging to the same semantic category as the target object but showing a different exemplar, different rotation and scaling. We expected subjects to indicate that the exemplar from the same category was more similar to the target than the non-return fixation locations. We set an exclusion criteria for subjects that made more than 3 errors in the control trials, but all 20 subjects satisfied these two controls and none were excluded from the analyses.

## Computational model to predict return fixations

We first provide a high-level intuitive outline of the proposed computational model, followed by a full description of the implementation details (Fig 7). The model is based on the previously published architecture for invariant visual search (IVSN [8]). The current model incorporates many modifications, most notably the prediction of eye movements during free viewing conditions when there is no target to search for, the incorporation of multiple maps discussed below, and finite inhibition-of-return.

The output of the model is a sequence of fixations. During visual search tasks, there are two inputs to the model: the target image ($I_T$) and the search image ($I_S$). During free viewing tasks, there is only a single stimulus input ($I_S$). The model posits an attention map $M_{f,t}$ at each fixation time $t$ by integrating four components: a bottom-up saliency map $M_{sal}$, a target visual feature similarity map $M_{sim}$, a saccade prior map $M_{sac,t}$ dependent on the previous fixation location, and a visual working memory map $M_{mem,t}$ dependent on the location of previous fixations (Fig 7).

In the visual search tasks, both the target image ($I_T$) and the search image ($I_S$) are processed through the same deep convolutional neural network, which aims to mimic the transformation of pixel-like inputs through the ventral visual cortex [74–76]. The target feature similarity map ($M_{sim}$) indicates the similarity between the target image and each location of the search image.

$M_{sim}$ is computed using the same procedure described previously [8]. Briefly, feature information from the top level of the visual hierarchy provides top-down modulation, based on the target high-level features, on the activation responses to the search image. The target feature similarity map depends exclusively on $I_T$ and $I_S$ and does not change with each fixation. During the free viewing tasks, there is no target image, and we therefore remove the top-down modulation, and use the same deep convolutional neural network to extract the high-level features of the stimulus $I_S$, aggregating all the feature maps into one saliency map $M_{sal}$.

Inhibition-of-return refers to the observation that previously fixated locations tend to be inhibited [17]. Many models of visual search, including the initial version of IVSN [8], assume infinite inhibition-of-return. Under these conditions, there cannot be any return fixations since these models remember perfectly the previously visited locations and never look back. Modeling return fixations requires a finite memory. Many studies [31, 34–38] have capitalized on return fixations to study visual working memory. Here we introduced a memory decay function to keep track of previous visited locations and constantly update the visual working memory map $M_{mem,t}$ over all past fixations from 1 to $t$−1 (S10(A) Fig). $M_{mem,t}$ depends only on the previous fixations and is independent of the content of $I_T$ or $I_S$.

Researchers have also shown that oculomotor biases constrain the saccade sizes (e.g., subjects are more likely to make two 10 dva saccades than one 20 dva saccade, [57]). Together with eccentricity-dependent sampling [61], this oculomotor constraint can also impact the frequency of return fixations [39]. To take saccade size constraints into account, the model incorporated a saccade prior distribution map $M_{sac,t}$. $M_{sac,t}$ depends only on the previous fixation $t$ −1 and is independent of the content of $I_T$ or $I_S$.

The final attention map $M_{f,t}$ integrates $M_{sim}$, $M_{sal}$, $M_{mem,t}$ and $M_{sac,t}$. A winner-take-all mechanism selects the maximum local activity as the location for the next fixation at $t + 1$. During visual search tasks, if the model recognizes the target at the current fixation location, the search stops. Otherwise, the maps are updated and the model produces a new fixation. During the free viewing tasks, the model stops when it reaches the average number of fixations made by humans or monkeys in the corresponding datasets.

The model was always presented with the exact same images that were shown to the subjects in all the tasks. We focus here on modeling the results for only the first six experiments on static images for several reasons. First, the model does not have any mechanisms for processing motion information or integrate temporal information across video frames. Second, the model does not have any mechanism to incorporate specific task information such as following a recipe in the cooking task. Third, a model of the egocentric videos would require constructing a memory map in 3D.

**Target feature similarity map.** The computation of the target feature similarity map $M_{sim}$ follows the IVSN model in reference [8]. Of note, this is a zero-shot model which does not require training on any eye movement data. We describe the computation of $M_{sim}$ briefly here and refer the reader to reference [8] for further details. The "ventral visual cortex" module builds upon the basic bottom-up architecture for visual recognition [52, 74, 75, 77–79]. We used a deep feed-forward network, implemented in VGG16 [52], pre-trained for image classification on the 2012 version of the ImageNet dataset [80]. The same set of weights, that is, the same network, is used to process the target image $I_T$ and the search image $I_S$. The output of the ventral visual cortex module is given by the activations at the top-level (Layer 31 in VGG16, $\phi_{31}(I_T, W)$, and the layer before that (Layer 30 in VGG16), $\phi_{30}(I_S, W)$, in response to the target image and search image, respectively. The top level activation is stored in a "pre-frontal cortex" module. We use the activations in layer 31 in response to the target image to provide top-down modulation to layer 30's response to the search image (Fig 7 and S9(A) Fig). This modulation is achieved by convolving the representation of the target with the representation of the

search image before max-pooling:

$$M_{sim} = f(\phi_{31}(I_T, W), \phi_{30}(I_S, W)) \tag{1}$$

where $f(\cdot)$ is the target modulation function defined as a 2D convolution operation with kernel $\phi_{31}(I_T, W)$ on the search feature map $\phi_{30}(I_S, W)$.

**Saliency map.** The saliency map $M_{sal}$ is computed using the same ventral visual cortex module (without any weight changes or retraining). We obtained the activations of size $C_{30} \times W_{30} \times H_{30}$ at the top-level (Layer 30 in VGG16) where $C_{30}$ is the number of channels, $W_{30}$ and $H_{30}$ are the width and height respectively. We take the average over all channels. In other words, $\phi_{30}(I_S, W)$ gets uniformly modulated by an all-ones matrix $J_{C_l \times 1 \times 1}$ of size $C_l \times 1 \times 1$.

$$M_{sal} = f(J_{C_l \times 1 \times 1}, \phi_{30}(I_S, W)) \tag{2}$$

**Saccade prior map.** Humans and monkeys make relatively small saccades, probably due to a combination of eccentricity-dependent sampling [7, 61] and oculomotor constraints [8, 39]. We used the empirical distribution of saccade sizes to constrain the saccade sizes for the model. Specifically, we plotted the saccade size distribution of the subjects on the corresponding datasets and interpolated to create a 2D map $M_{sac,t}$ centered at the $t$th fixation. S10(B) Fig plots the empirical saccade size distributions of all fixations over all trials and subjects for each dataset and their corresponding 2D saccade maps when the fixation is at the center. The saccade prior map is updated after each fixation. S11(E) Fig shows example visualizations of saccade priors over fixations.

**Memory decay map.** Humans and monkeys have limited memory capacity and finite inhibition of return [17, 58, 59]. We added a finite memory module to the model where the 2D memory map $M_{mem,\tilde{t}}$ at the $\tilde{t}$th fixation keeps track of memories at all the past fixation locations $\{(x_1, y_1), (x_2, y_2), ..., (x_t, y_t), ..., (x_{\tilde{t}}, y_{\tilde{t}})\}$. From the $\tilde{t}$th back to the 1st fixation locations, the memory value $a_t$ at the $t$th fixation location gets degraded using the following memory decay function:

$$a_t = \begin{cases} \alpha^{\tilde{t}-t}, & \text{if } \alpha^{\tilde{t}-t} \geq \beta \\ \beta, & \text{otherwise} \end{cases} \tag{3}$$

where we set memory decay parameter $\alpha = 0.92$ and clipping threshold $\beta = 0.5$. S10(A) Fig shows the plot of memory value $a_t$ as a function of fixation number $t$ when $\tilde{t} = 15$. The model's memory decays for the most recent fixations and maintains a low memory level for the rest of past fixations. To avoid sparseness of the 2D memory map $A_{mem,t}$ for the $t$th fixation at $(x_t, y_t)$, we applied Gaussian filtering centered at that fixation location:

$$A_{mem,t}(x_t, y_t) = a_t \exp^{-\frac{(x-x_t)^2 + (y-y_t)^2}{2\sigma^2}} \tag{4}$$

where $\sigma$ is the standard deviation of the Gaussian. The value of $\sigma$ controls how much the model remembers adjacent pixels centered around fixation location $(x_t, y_t)$. We set $\sigma = 0.08$ on object arrays (Visual Search 1) and $\sigma = 0.02$ in the other datasets. The different $\sigma$ is because in object arrays, each object stands alone, and the choice of the Gaussian memory mask is large enough to cover the complete object on the arrays. To avoid overfitting, we optimized $\alpha$, $\beta$ and $\sigma$ *only* on the Visual Search 2 task, and fixed those parameters for the rest of the tasks.

After updating $A_{mem,t}(x_t, y_t)$ for each fixation $t$, the model predicts the final memory map $M_{mem,\tilde{t}}$ by taking the largest memory value across $A_{mem,t}(x_t, y_t)$ for all previous fixation

locations $t \in \{1, 2, ..., t, ..., \tilde{t}\}$. S11(F) Fig shows visualization examples of the memory map $M_{mem,\tilde{t}}$. When there is a return fixation, by taking the largest memory value across $A_{mem,t}(x_t, y_t)$, the memory value at the revisited location overwrites the decayed memory value at the "to-be-revisited" location.

**Integration of feature maps.** The model predicts the final attention map $M_{f,t}$ by taking the weighted linear combination of $M_{sim}$, $M_{sal}$, $M_{mem,t}$ and $Msac$, $t$, after normalizing them to $[0, 1]$ (S11 Fig).

$$M_{f,t} = w_{mem}M_{mem,t} + w_{sac}M_{sac,t} + w_{sim}M_{sim} + w_{sal}M_{sal} \tag{5}$$

In the visual search tasks, $w_{sim} = 1$ and $w_{sal} = 0$ and in the free viewing tasks $w_{sim} = 0$ and $w_{sal} = 1$. We fit the 2 weights $w_{mem}$ and $w_{sac}$ to approximate the return fixation properties *only* on the Visual Search 2 experiment with natural images. We used the following weights: $w_{mem} = -0.93$, $w_{sac} = 0.2346$. These weights are fixed throughout the experiments and do not depend on $t$. The model takes the maximum in the attention map $M_{f,t}$ as the location of the $t + 1$-th fixation.

**Object recognition.** In the visual search tasks, given a fixation location, the model needs to decide whether the target was found or not (in a similar way that humans need to decide whether they found the target after moving their eyes to a new location). The model performs visual recognition to decide whether the target is present at the fixated location. We used a simplified visual recognition mechanism consisting of four steps: (1) we cropped a patch of 1 dva centered at the current fixation; (2) we used the same deep feed-forward architecture described above to extract the activations in the last classification layer of VGG16 in response to the cropped patch; (3) we similarly extracted the activation in the last classification layer of VGG16 in response to the target image $I_T$, and (4) we computed the cosine similarity distance between the activations for the image patch and the target image.

We computed $M_{recog}$ for each location in the image. At each fixation location, the model retrieved the cosine similarity distance in $M_{recog}$. We empirically set a hard threshold for cosine similarity distance (0.5 for object arrays and Waldo images; 0.3 for natural images). If the distance between the current fixation patch and the target image is below the threshold, the model decides that the target is found and the search trial stops; otherwise, the model continues the visual search process by updating the four maps and the overall attention map (Eq 5).

In the free viewing tasks, there is no target image to recognize. Instead, we stop the model after it generated $N_c$ fixations, where $N_c$ is the average number of fixations by humans or monkeys in the corresponding dataset.

**Ablated models.** We assess the model components by testing the following ablated models in all the six experiments.

**Ablated target feature similarity map.** In the three visual search experiments, we removed the target feature similarity map by setting $w_{sim}$ to be zero in Eq 5. Since the ablated model now predicts image feature-agnostic attention maps for all the images, selecting the maximum of the attention map as the location for the next fixation would result in generating the same sequence of fixations for all the images. Thus, we introduced stochasticity to the winner-take-all mechanism by sampling the next fixation location from the attention map $M_{f,t}$ so that the model outputs different sequences of fixations for different images.

**Ablated saliency map.** In the three free viewing experiments, we removed the saliency map by setting $w_{sal}$ to be zero in Eq 5. Similar as **Ablated target feature similarity map**, we introduced stochasticity to randomly sample the next fixation location from $M_{f,t}$.

**Ablated saccade prior map.** We enforced $w_{sac}$ to be zero in Eq 5. The winner-take-all mechanism selects the maximum of the attention map as the location for the next fixation.

**Ablated memory decay map.** This model considered the possibility that prior location does not effectively get added into the memory. This defective memory module was implemented by randomly assigning a value between [-1,0] to $w_{mem}$ in Eq 5.

**Null model.** We compared the model performance against a null model that made random eye movements. Similar to reference [39], the null model is memoryless: it does not have any history dependency during prediction of the next fixation location. The only constraint that the null model has is the saccade size, ensuring that the null model can also make return fixations by selecting random locations in the vicinity of the current fixation. Thus, the null model randomly samples the next fixation location from the saccade amplitude distribution $M_{sac,t}$. We used the same stopping criteria for the null model as the one described in the previous paragraph. We ran simulations generating at least 25,000 random sequences of fixations and reported their proportion of return fixations for all datasets in Fig 3, S1 Fig and S3 Fig. These random sequences of fixations have the same length as the average number of fixations per trial for each dataset. Similarly, the number of return fixations also impacts the entropy value (S4 Fig). In order to calculate the chance level for the between-subject consistency analyses, we used the number of return fixations collected from all subjects in each trial (that is, each image) to randomly generate an equal number of random return fixations and computed the entropy for these random return fixations. We repeated this process 100 times for every trial per dataset.

In the null model described above, we do not take bottom-up saliency map into account. An alternative null model would be to incorporate $M_{sac,t}$, $M_{sim}$ and $M_{sal}$ with $M_{mem,t}$ removed. Without $M_{mem,t}$, $M_{sal}$ would dominate the final attention map $M_{f,t}$. Hence, such an alternative null model would fixate back and forth between the maximum and the second local maximum on alternating $M_{f,t}$ and $M_{f,t+1}$. This would result in strong exploitation of those two locations without any exploration.

## Data analyses

**Fixation extraction and calibration.** In the visual search tasks, we used the fixations from the previous work [8]. In the human free viewing task, we used the fixation clustering function from [81], implemented in MATLAB. During all the human eyetracking experiments on static images, if a fixation was not detected during the initial fixation window in each trial, the experimenter re-calibrated the eye tracker.

In the Monkey Free Viewing 1 task, there were 3–5 re-calibrations. We minimized the number of times for checking re-calibrations in order to maintain the monkeys' attention. The checking process was only activated if the monkeys failed 3 times or more to complete a trial. We used two eyetrackers (four monkeys using ISCAN, 60 Hz sampling rate and <1 dva resolution, and two monkeys using Eyelink 1000 Plus, 1000 Hz sampling rate and <1 dva resolution) to record monkeys' eye positions. In the Monkey Free Viewing 2 task, calibration was conducted in the beginning of each session. We used the ISCAN eyetracker to record the monkeys' eye positions. We used the built-in Monkeylogic 2 graphics library [82] to monitor eye movements on MATLAB.

In the two egocentric datasets [48, 49], calibration was only performed once before each experiment. In a natural environment the resulting egocentric videos represent a combination of head pose and gaze position. Different from static images, both foreground and background objects move with respect to the egocentric coordinate. This implies that the coordinates of a fixated object on the current video frame might be shifted with respect to the next frame or even disappear in the next frame due to abrupt large head motions. To avoid the complexities of using optic flow to track coordinates across frames, we simplified the analyses by

considering short segments with minimal head motion. We uniformly split the long egocentric videos into short 5-second video clips. To approximate head motion, we calculated the Euclidean distance between the first frame and the last frame of the video clip at the pixel level and normalized the Euclidean distance by the total number of pixels on each frame (S2 Fig). We only considered video clips if the normalized Euclidean distance between the first frame and the last frame was ≤ 0.4.

The field of view of the camera capturing the egocentric videos was limited to 46 × 60 dva while the field of view for human eyes is at least 120 dva. Thus, we could have cases when the eye tracking data goes beyond the size of the video frame, leading to missing fixations in some video frames. Missing eye tracking data could also arise as a consequence of large head rotations. The coordination between head and gaze movements results in an early movement of eye gazes to the anticipated direction of what subjects intend to look at before the head rotates such that subjects can re-position the object of interest in the center of the field of view [69]. Therefore, in addition to the constraint based on the normalized Euclidian distance, we also discarded from analyses those video clips with more than 14 consecutive frames with missing fixations. Fourteen consecutive frames at 24 frames per second corresponds to about 3 fixations. For the rest of the video clips with fewer missing fixations, we performed linear interpolation to estimate eye positions in frames with missing data. After all these pre-filtering steps, we ended up with 8,468 video clips in the cooking egocentric videos (Fig 2G) and 1,186 clips in the visual search egocentric videos (Fig 2H). Despite these efforts to remove large head motion, we could still have video clips with small head movements, which would lead to inaccurate analyses of return fixations. Therefore, for the egocentric videos, we relaxed the threshold in the definition of return fixations to 1.5 dva overlap, instead of 1 dva overlap as used in all the other datasets.

**Evaluation of object recognition during visual search.**  During visual search, subjects can fixate on the target object without realizing that they have found the target and continue searching (there are multiple examples of this phenomenon and discussion in reference [8]). We refer to these cases of missing the targets as "false negatives" in visual recognition (S21 Fig). To compute the false negative rate for humans, we counted the total number of fixations on the targets without mouse clicks and divided it by the total number of fixations for all the trials (S21 Fig).

Conversely, there could also be cases when humans are not looking at the target but rather they are fixating on a distractor; yet, they misclassify the distractor as the target by clicking the mouse at the wrong location. We refer to these failure cases as "false positives". We computed the false positive rate as the number of trials when false clicks happen within that trial, normalized by the total number of trials (S21 Fig).

In the object array experiments, subjects were not asked to click the target location with the mouse and therefore we cannot compute false positives or false negatives. Thus, we only reported false positive and false negative rates in the Visual Search 2 and 3 datasets (natural images and Waldo). The model similarly makes false positives and false negatives, also reported in (S21 Fig).

**Evaluation of return fixation properties.**  *Definition of return fixations:* A fixation location was considered to be a *return fixation* if it was within one degree of visual angle (dva) of a previous fixation location (Figs 1 and 3I). One dva is approximately the resolution of the eye tracking data in the experiments reported here. The original fixation is referred to as a *to-be-revisited* location. All other fixations are referred to as *non-return* locations. This definition is consistent with criteria previously used in the literature (e.g., [26]). In the egocentric video datasets (Fig 2G–2H), we relaxed the degree of overlap to 1.5 dva to account for the alignment imprecision introduced by potential head movements.

Throughout the manuscript we used 1 dva as the criterion threshold to define a location as a return fixation and 1.5 dva in egocentric videos (Figs 1 and 3I). To assess the robustness of the results to this definition, we repeated all the analyses using a threshold of 2 dva, 3 dva, or 4 dva (S22 Fig). The qualitative conclusions about the patterns of return fixations in humans, monkeys, and the model are not changed with different thresholds. The quantitative values do change; for example, trivially, the proportion of return fixations increases dramatically as the radius is enlarged. Moreover, there is an even better model-experiment consistency when using larger thresholds (S22 Fig). We keep the definition of 1 dva because this is approximately the resolution of the eye tracker, therefore providing a good reference to indicate that two fixations actually correspond to the same location.

*Proportion of return fixations:* The proportion of return fixations is defined as the number of return fixations normalized by the total number of fixations in each trial. Fig 3A–3H reports the proportion of return fixations for every subject and Fig 8B reports the proportion of return fixations for the model. In the visual search tasks, we further divided the return fixations based on whether they landed on the target objects or non-target locations (Fig 6B). The proportion of return fixations at target locations was calculated as the number of on-target return fixations divided by the total number of on-target fixations per trial. Similarly, we calculated the proportion of non-target return fixations. We omitted the division of on-target and non-target return fixations on the egocentric visual search dataset because of the lack of annotations of target object locations in each video frame.

*Return offset:* The *return offset* was defined as the number of intervening fixations in between a to-be-revisited location and a return fixation. For example, in Fig 3I the return offset between to-be-revisited location 3 and return fixation 6 is 2. Fig 3J shows the distributions of return offsets for all experiments and Fig 8C shows the corresponding distributions for the model.

*Saliency:* To study whether return fixations correlate with saliency, we evaluated saliency maps using Graph-based Visual Saliency (GBVS), which is a bottom-up saliency prediction algorithm in computer vision using low-level visual features, such as color, orientation, and contrast [51]. We computed the average of all saliency values for each fixation patch, defined as a squared region covering one dva and centered at the fixation location. Fig 6A shows saliency for all the experiments and Fig 9C shows saliency for the model.

In the case of egocentric videos, it is not justifiable to establish a one-to-one mapping from an extracted fixation (lasting approximately 250 ms) to a single video frame (about 42 ms, 24Hz video frame rate). The fact that one fixaiton involves many frames implies the need for further assumptions to calculate the saliency value for a fixation. Assuming that we have little head motion for the selected video clips, we approximated the saliency value for a fixation by projecting all the fixations within a video clip back to the last frame, and computed saliency using GBVS on the last frame.

*Mouse clicks:* In the Visual Search 2 and Visual Search 3 experiments, subjects were asked to use the mouse to click on the location of the target. We asked whether there were additional return fixations at the end of each trial which were related to this testing procedure. For example, subjects may find the target, then look for the mouse position, and then make a second saccade to the target. We conducted two additional analyses to evaluate this possibility. First, if subjects look back for the mouse pointer, the fixation preceding the return fixations should overlap with the mouse pointer location. Subjects were instructed not to move the mouse during visual search, except when the target was found at the end of the trial. We computed the proportion of fixations preceding return fixations that overlapped with the mouse pointer location with respect to the total number of return fixations in both experiments. There were only 1.5% and 1.3% of preceding return fixations overlapping with mouse pointer locations in

the Visual Search 2 and 3 experiments, respectively. Second, to further quantify the effect of looking for mouse pointers, we examined the first 6 fixations in each trial (S3 Fig). In both experiments, it took subjects much more than 6 fixations to find the target [8]. During the initial 6 fixations, it is unlikely that the subjects were looking for the mouse pointer. During the first 6 fixations, the proportion of return fixations was significantly above chance in the Visual Search 2 experiment, but not in the Visual Search 3 experiment. In sum, the presence of return fixations in the visual search experiment 1 (where subjects did not use the mouse), combined with the presence of return fixations during the first 6 fixations in the Visual Search 2 experiment, and together with the low fraction of return fixations where subjects look for the mouse pointer before making a return fixation, suggest that the majority of return fixations cannot be ascribed to subjects searching for the mouse pointer.

**Between-subject consistency in return fixation locations.** To evaluate between-subject consistency (S4 Fig and S5 Fig), we performed the following steps: (1) we mapped all return fixations from all subjects on each image to a uniform 2D grid of size 32 by 40 (S4(B) Fig); (2) we computed the proportion of subjects that showed a return fixation at location $l$, with $l = 1, \ldots, 32 \times 40$; (3) computed the entropy $H$ for this distribution over $L$ locations using the following equation:

$$H(L) = -\sum_{l=1}^{32 \times 40} p_l \log(p_l) \tag{6}$$

For the entropy calculation, we took the following measures to avoid singularity and take into account sparsity. First, to avoid singularity where the probability value is 0, we set the 0 probability values to be $10^{-10}$. Second, to take into account the sparsity of the fixation positions, we applied Gaussian blur on the 2D histogram to soften the probabilistic distribution.

An extreme case of perfect consistency would lead a probability of 1 at a given location and 0 elsewhere, leading to minimal entropy, whereas a complete lack of consistency would lead to an approximately uniform probability distribution except for random overlaps, resulting in high entropy. We omitted this analysis in the egocentric video datasets because the field of view in each frame could be different across subjects, making comparisons difficult to interpret.

**Model-subject consistency in return fixation locations.** To evaluate the model-subject consistency in S16 Fig—S20 Fig, we introduced the Similarity Index (SI) for each property per experiment:

$$SI = 1 - (S - M)/(S + M) \tag{7}$$

where $S$ and $M$ are the property values of the average subject (S) and the model (M), respectively. The property value refers to the proportion of return fixations (S16 Fig), return offset (S17 Fig), saccade size (S18 Fig), saliency values (S20 Fig), and turning angles (S19 Fig). If a model follows the return fixation patterns of primates for a given experiment, it will have a high Similarity Index.

**Statistical analyses.** In each experiment, we calculated the average for each subject and performed statistical analyses across subjects using two-tailed t-tests (Figs 3, 4 and 6, S1 Fig and S3 Fig). For the model, two-tailed t-tests were computed across fixations for all properties, except for the proportion of return fixations in which statistical significance was determined using bootstrapping (resampling with replacement) with 1000 resamples. These methods were used to evaluate differences between conditions and also to compare experimental results against chance levels.

## Supporting information

**S1 Fig. Proportion of fixations that revisit the same location twice.**
(TIF)

**S2 Fig. Extraction of fixations on egocentric videos.**
(TIF)

**S3 Fig. Proportion of return fixations among the first six fixations.**
(TIF)

**S4 Fig. Return fixations are consistent across subjects.**
(TIF)

**S5 Fig. Example of return fixation consistency across subjects.**
(PDF)

**S6 Fig. Distribution of saccade sizes.**
(TIF)

**S7 Fig. Saccade sizes and angles for primates.**
(TIF)

**S8 Fig. Return and non-return fixation locations.**
(PDF)

**S9 Fig. Calculation of bottom-up saliency map, target similarity map, and recognition map in the computational model.**
(TIF)

**S10 Fig. Memory decay function and 2D empirical distribution of saccade sizes.**
(TIF)

**S11 Fig. Visualization examples of attention maps for the model.**
(TIF)

**S12 Fig. Example return fixations in model predictions.**
(TIF)

**S13 Fig. The model makes more return fixations at target locations than non-target locations in visual search.**
(TIF)

**S14 Fig. The model makes shorter saccades at return fixations than non-return fixations.**
(TIF)

**S15 Fig. Saccade sizes and turning angles for the model.**
(TIF)

**S16 Fig. Model ablations reveal critical model components.**
(TIF)

**S17 Fig. Effect of model ablations on return offset.**
(TIF)

**S18 Fig. Effect of model ablations on saccade size distribution.**
(TIF)

**S19 Fig. Effect of model ablations on turning angles.**
(TIF)

**S20 Fig. Effect of model ablations on saliency at fixated locations.**
(TIF)

**S21 Fig. False negative and positive rates for humans and the model in Visual Search 2 on natural images (A) and in Visual Search 3 on Waldo images (B).**
(TIF)

**S22 Fig. Effect of threshold to define return fixations.**
(PDF)

**S23 Fig. Primates and our model tend to make longer saccades with larger turning angles.**
(TIF)

**S24 Fig. Return fixation durations are longer for target compared to non-target locations.**
(TIF)

## Acknowledgments

We thank Jeremy Wolfe for helpful discussions and advice, Pranav Misra for help with the datasets, Yin Li, Miao Liu and Keng Teck Ma for helping with the egocentric video datasets. We thank the BPRC, Rijswijk, The Netherlands, for providing images for our study.

## Author Contributions

**Conceptualization:** Mengmi Zhang, Will Xiao, Carlos Ponce, Gabriel Kreiman.

**Data curation:** Mengmi Zhang, Will Xiao, Carlos Ponce.

**Formal analysis:** Mengmi Zhang, Marcelo Armendariz.

**Funding acquisition:** Mengmi Zhang, Gabriel Kreiman.

**Investigation:** Mengmi Zhang, Marcelo Armendariz, Will Xiao, Olivia Rose, Katarina Bendtz.

**Methodology:** Mengmi Zhang.

**Project administration:** Mengmi Zhang, Gabriel Kreiman.

**Resources:** Gabriel Kreiman.

**Software:** Mengmi Zhang.

**Supervision:** Margaret Livingstone, Carlos Ponce, Gabriel Kreiman.

**Validation:** Mengmi Zhang.

**Visualization:** Mengmi Zhang, Marcelo Armendariz.

**Writing – original draft:** Mengmi Zhang, Gabriel Kreiman.

**Writing – review & editing:** Mengmi Zhang, Marcelo Armendariz, Will Xiao, Olivia Rose, Margaret Livingstone, Carlos Ponce, Gabriel Kreiman.

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
