## [Decision Letter · Decision Letter 0]

30 Jan 2022

Dear Prof. Kreiman,

Thank you very much for submitting your manuscript "Look twice: balancing exploration and exploitation in eye movements" for consideration at PLOS Computational Biology.

As with all papers reviewed by the journal, your manuscript was reviewed by members of the editorial board and by several independent reviewers. In light of the reviews (below this email), we would like to invite the resubmission of a very significantly-revised version that takes into account the reviewers' comments. I note that we see this in light of the reviewers comments as substantial effort is required as concerns extend as far as the presence of sufficient innovation and relevance.

We cannot make any decision about publication until we have seen the revised manuscript and your response to the reviewers' comments. Your revised manuscript is also likely to be sent to reviewers for further evaluation.

Sincerely,

Aldo A Faisal

Associate Editor

PLOS Computational Biology

Wolfgang Einhäuser

Deputy Editor

PLOS Computational Biology

Reviewer's Responses to Questions

**Comments to the Authors:**

Reviewer #1: Zhang et al. 2021

In this report, Zhang et al. provide both a detailed empirical description of and computational model of return fixations in humans and monkeys across a variety of visual tasks. The authors show that return fixations are quite common and not unique to specific stimulus configuration or task instructions. They show that a model of visual processing, which incorporates stimulus salience, task relevance of features, along with working memory and constraints on eye movements can produce fixation sequences that also predict that return fixations would also occur. Together, the authors suggest that return fixations can be considered in a framework related to the balance between exploration and exploitation.

In general, I found the paper to be well written and to provide a comprehensive study of the phenomenon of "return fixations". To the extent that the paper's aim is to demonstrate the ubiquity of these I think it succeeds. I do have concerns that in the present form that the modeling component does not add as much to the story as it could. In particular, despite the clear methodological description of the model itself in the latter parts of the manuscript, the impact of the model in terms of providing key insights or conceptual advances is modest. Tied to this concern is the potentially interesting connection to "explore and exploit". But here again, I expected the model to make particular predictions about this balance. Instead, from my reading, the model is essentially used to show that a combination of constraints (admittedly including some elements not present in basic saliency models) would predict that return fixations should occur. But again there seem to be no real key insights about what the model predicts about behavior more generally. I think the paper falls somewhere between an empirical paper and a computational paper, but doesn't effectively merge these as well as it potentially could.

From the Introduction, I think it would be helpful to include some "significance" to the last paragraph. The authors write: "We show that both monkeys and humans make frequent return fixations with few intervening saccades, especially but not exclusively during visual search..." And then they say that they want to "gain insights into the mechanisms", they look to their refined model. But what are the key takeaway insights? Are there ways to characterize optimal return fixations proportions based on parameters of the model or manipulation the likelihood of their occurrence? If subjects were guided to slow down their eye movements, e.g., would this reduce the proportion of return fixations observed?

As an example of a question that seems appropriate for the model to address, and for which the empirical data might provide guidance, would be the degree to which different parameters could effect the likelihood of return saccades. In particular, it seems like the memory component in the model is one place the authors might consider including the possibility that the prior location doesn't effectively get added into the memory. As opposed to a global decay, is it not plausible that specific locations fail to get added (perhaps as a function of something else, like stimulus complexity or fixation duration). In any case, while I can't say I have specific hypotheses about the relative contributions of the target similarity factor or the memory decay, it would make the inclusion of the model stronger if it did more than predict that inclusion of the extra terms led to return saccades.

Regarding the model fit, the assessment of the quality of the model fit seems a bit arbitrary. For example, in Figure S10 the caption reads "The distribution of saccade sizes in the model matches the one from humans and monkeys", but the first in 2-6 don't seem good at all (and the distribution in 1 is highly unusual?). In any case, the loose use of the term "matches" would seem to undermine the inclusion of the model as anything more than a vague proof of concept.

I think the comparison of return fixations across task is a positive, but I'm a little confused about how to think about return fixations in a dynamic video? As an extreme case, would we be surprised to show that drivers make return fixations when driving (as they repeated look at the controls on the dash?). The cooking video would seem to fall into this camp.

There is also a missed opportunity to consider the dynamics of this process, which seems so critical to this problem (this is noted by the authors on page 9). In the Introduction, the suggestion is made that "return fixations could constitute a useful strategy during visual search, especially in difficult tasks" (p. 3) but it's not clear that "useful strategy" is the best term here? It seems like the issue is more of a mismatch between fixation duration and the inability to extract relevant information in time, so the system is required to overcome these misses? It may be a semantic thing, but does seem to highlight the possibility of exploring how the proportion of return saccades depends on saccade rate.

Specific comments

Analysis: It's not clear what the authors are actually comparing in the reported statistical tests (I do see the short description on page 21). For example, on page 3: "the proportion of return fixations was higher than expected by a null model implementing random eye movements while respecting the distribution of saccade sizes (p < 10�15, two-tailed t-test, t < �16, df = 1, 598, Methods)". Where are these degrees of freedom coming from and is a t-test really appropriate here? For these analyses it seems like all fixations are just collated into a single large analysis. I would have thought a more appropriate analysis would have been to use each subject's average as a datapoint? To that end, the bar charts in Fig 3 (and S1, S2, and S4) could be composed on single points from each subject, which would provide a more robust analysis of these data, demonstrating that all subjects make return saccades (which I suspect is the case).

Methods: why is the eye tracker introduced in the "Human free viewing" section and not the "Visual search on static images" above?

Methods: The choice of 1dva for defining a return saccade seems reasonable, but as this entire paper depends on this parameter, it would have been nice to see that the entire analysis pipeline would lead to the same conclusions with different choices for this value.

p. 8 "oculomotor" is misspelled "occulomotor"

Reviewer #2: The manuscript describes experimental and modeling studies on return fixations during eight image (or movie) exploration/search tasks.

The choice between fixating new locations and returning to an old fixation location is conceptualized as a trade-off between

exploration of new information and exploitation of previously encountered, but potentially uncertain information. While this trade-off

makes intutitve sense, the manuscript does not present compelling evidence for it. Instead, fixation locations in the different tasks are

compared to a null model of random fixations. It is found that fixations are not random, which is unsurprising. Furthermore, even when there is

no explicit incentive to return to a previously fixated location in the free-viewing tasks, return fixations are still abundant (page 3, 'return fixations are ubiquitous'), which seems to contradict the exploration/exploitation trade-off explanation. Further results are the consistency of the return fixations across subjects, the more frequent returns to salient locations, and locations similar to the target in the search tasks.

These observations inspire an image-computational model, which is the most interesting and novel aspect of the manuscript in my opinion. Feature maps are computed with a pre-trained 'vental stream' model (VGG-16). These feature maps are used to compute a saliency map and a similarity map (in case of the search tasks). Furthermore, to incorporate IOR a memory map is added to the model, which makes the return to a very recently added location unlikely. Additionally, a 'saccade size constrain' map ensures that the generated saccades match observations without having to implement a complete oculomotor system. While there are plenty of attention models based on saliency and task-relevant information, the addition of a memory map and saccadic size constraints constitutes a novelty, whose importance to the field could be appreciated even more if it was more rigorously evaluated.

Major comments:

- The idea of viewing novel vs. return fixations as 'exploration vs. exploitation' is interesting, but I was not able to find compelling evidence for this distinction in the experiments or the model. On the experimental side, I would have expected reward manipulations to demonstrate that return fixations are indeed exploitative. On the modeling side, a reinforcement learning component to the model might have helped in substantiating this claim. I would like to ask the authors to add data/models to this end, or reconsider the metaphor.

- The model deserves more quantitative evaluation before it can be considered for publication. As the authors themselves point out, more relevant null models than a random fixation model could be easily constructed from the computational model, e.g. by removing the memory map and/or the target similarity map. Quantitative model comparison scores for relevant aspects of the paper (Hit rates, return offsets etc.) between humans/monkey and model should then be computed, to demonstrate the benefit of the proposed model architecture over standard saliency models.

Reviewer #3: Review on “Look twice: balancing exploration and exploitation in eye movements” by Zhang et al.

Overall evaluation

The manuscript investigates return fixations in scene viewing and visual search across tasks and species. Results are interpreted from the perspective of exploration vs. exploitation. In the modeling part, an activation-map based framework is proposed which uses concepts of attention, memory decay, and inhibition of return. While implemented as a numerical model, the current modeling work is on a qualitative level to demonstrate basic properties of return fixations. I am in favor of the manuscript which clearly adds to the literature, the innovation, however, is rather limited when compared to previous experimental as well as modeling work. Critically, the reader might want to see more analysis of data and model as well as statistical model inference in a computational biology journal.

Comments

1) Abstract: The most important topic of the abstract, return fixations, has been studied intensely before, across series of experiments. What should be highlighted in the abstract are the new aspects, i.e., the experiments across tasks and species. Also, the computational model rather preliminary in its current form and some aspects lag behind already published work.

2) p. 2: Attention and IOR maps: Dynamical activation-based maps have been proposed as a biologically-inspired modeling framework before (SceneWalk model), published first by Engbert et al. (2015, Journal Vision), with likelihood-based parameter inference by Schütt et al. (2017, Psychological Review), and most recently in an advanced version with attentional processes around the time of saccade and fully Bayesian inference by Schwetlick et al. (2020, Communications Biology). I understand that there are differences between theses previous works and the current work, however, these previous papers should be discussed and compare with the current model. Also, I feel that a numerically implemented model w/o statistical parameter inference might be difficult to publish in a computational biology journal.

3) p. 2: Exploration vs. exploitation: The author should make clear that the paper by Malem-Shinitski et al. [11] tried to build a model within the same framework.

4) p.2: In the last paragraph, the author discuss the new aspects of the manuscript, related to generalizations across tasks and species. I recommend to extend this part throughout the manuscript, but I have questions related to data quality and number of experimental subjects below.

5) Results: A general comment on the Results is that little citations are given here; the author condensed almost all references to the introduction, however, in such a format, the details of the previous studies cannot be related to the specific results discussed here. So, while I am aware that Results is not Discussion, the reader needs more comparisons to previous work in the Results as well to appreciate the findings.

6) p. 3: Return fixations are ubiquitous: The basic finding reported in this part is not new, which should be explained to the reader. The comparison between species and tasks should be addressed more deeply.

7) p. 5: Subjects promptly revisit return fixations: I am not convinced about the interindividual differences results, which should be addressed in a statistically adequate way (e.g., linear-mixed effects modeling with task and subject as random factors to provide statistical evidence for stability of individual viewing behaviors).

8) p. 5: The authors state that there was “no consistent relationship between the duration of the to-be-revisited fixation and non-return fixation.” One would see a bit more details in this direction. For example, Schwetlick et al. (2020, Communications Biology, Fig. 5 + 6) demonstrate that there is systematic relation between turn angles and saccade amplitude as well as fixation durations. This has implications for the modeling part as well (see my next point).

9) p. 7: A computational model: Because of the complex coupling between fixation durations and saccade amplitudes to saccadic turning angle, the proposed model should be investigated with respect to such statistics. Even if the author would like to stay at a qualitative level (which is OK with me), a model that cannot reproduced basic statistics such as the distribution of turning angle would not be acceptable when it is used as a qualitative tool to explain return fixations.

10) p. 8: The assumption that of constant acuity over the entire visual field is hard to accept in a biologically inspired model (see also last para of the Discussion on p. 11). Other papers (e.g., on the SceneWalk model) try to implement limited attention by a Gaussian attentional aperture, which naturally explains the saccade-size distributions. The arbitrary assumption of a fixed Gamma-type distributed should be avoided. Also, the author should visualize the saccade-size distribution of the model in comparison to the experimental data. I think the idea that oculomotor limitations [51, 52] provide the answer to this question is experimentally falsified, at least this is no longer a theoretically motivated argument.

11) p. 10: “… has very few parameters”: I agree, but this is precisely the reason why one would believe that statistical parameter inference is possible and should be investigated. Important questions are related to this point, e.g., the question whether the model could explain interindividual differences via parameter variation.

12) p. 11: “return fixations showed longer fixation durations”. See also Rothkegel et al. (2019, Scientific Reports, Fig. 7C). The bottom line from my comments is that the reader needs more details/visualizations/analyses on the data and model output to understand the results.

13) Methods: The presentation times were quite different across the different experiments. I would be surprised if this does not have an impact on the results. Please comment based on additional analyses.

14) p. 13: The use of a 30 Hz eye tracker with low spatial resolution is possible, however, role of smaller saccades will not be addressed in this case. Here, the main sequence and overall saccade amplitude distribution should be plotted and investigated to emphasize the reliability of the results.

15) p. 16: The Gaussian in Eq. (4) needs a negative exponent.

Typos:

- “occulomotor” throughout the manuscript

EOF

**Have the authors made all data and (if applicable) computational code underlying the findings in their manuscript fully available?**

Reviewer #1: Yes

Reviewer #2: Yes

Reviewer #3: Yes

PLOS authors have the option to publish the peer review history of their article (what does this mean?). If published, this will include your full peer review and any attached files.

Reviewer #1: No

Reviewer #2: No

Reviewer #3: No
---

## [Decision Letter · Decision Letter 1]

10 Aug 2022

Dear Prof. Kreiman,

Thank you very much for submitting your manuscript "Look Twice: A Generalist Computational Model Predicts Return Fixations across Tasks and Species" for consideration at PLOS Computational Biology. As with all papers reviewed by the journal, your manuscript was reviewed by members of the editorial board and by several independent reviewers. The reviewers appreciated the attention to an important topic. Based on the remaining reviewer and discussions here, we are likely to accept this manuscript for publication, providing that you modify the manuscript according to the review recommendations. We believe the clarifications requested by the reviews will help strengthen the manuscript and should be possible to turn around in reasonable time.

Sincerely,

Aldo A Faisal

Associate Editor

PLOS Computational Biology

Wolfgang Einhäuser

Deputy Editor

PLOS Computational Biology

[LINK]

Reviewer's Responses to Questions

**Comments to the Authors:**

Reviewer #1: In this revised ms, Zhang and colleagues have nicely addressed my primary concerns. In general I think the authors provide a broad set of convergent data highlighting properties of return fixations during visual search and scene viewing tasks in humans and monkeys. Their computational model, which naturally extends existing models for object processing and salience demonstrate that the inclusion of multiple maps with complementary functions (memory, salience, similarity) can lead to model performance that shares some of the properties found in real saccade data. In this way, the paper does a nice job of focusing on a problem that has not received extensive attention but which is part of real world vision. That said, the universal properties and the model configuration do not really provide novel or deep insights about the nature of return fixations. This is a result of the desire to look across multiple tasks that prevent detailed comparisons of controlled experimental manipulation and the desire to show that the simple combination of feature maps and the WTA architecture are enough to generate this pattern of behavior.

As indicated above, I am satisfied by the responses provided by the authors, as most of my comments were focused on considerations of aspects of the tasks or model that might be worth addressing in the introduction or discussion. These were addressed, and the "ablation" approach does address the relatively importance of the different modules. If there were one thing I think might still be missing in the general discussion of the model is the simplicity with which space is being treated. This makes sense for CNNs, where all the images can be magically aligned and the WTA operation is simple enough. Perhaps some comment about how this simplification has only the vaguest support from existing neuroscientific studies, as these maps appear to exist in some forms, but they are spread across different brain areas and the registration of space is hardly a solved problem.

While I think the paper make a valuable contribution and appreciate the effort put into the revision, I do have a couple of remaining issues/questions.

1. The authors suggest that they have set aside the explore/exploit viewpoint, but the idea is still introduced in 59ff and it's not really tied to anything that follows.

2. The ablation approach and the conclusions are highlighting on page 10, but all of the figures are buried in the supplementals. I'd think it might be worth somehow creating some figure that highlights the take home message from these and then referring to the supplemental data figures for the nitty gritty details? For the Null model, it is a little awkward that this couldn't be framed as a totally ablated model or put into this framework. I see the argument presented in 932ff, but it would seem more parsimonious to connect these.

3. There is also no main figure for the consistency analysis (181ff)?

4. 180 degree turns. It's not clear how this metric is not confounded by the "offset" measure. I still don't understand how a 1 offset return saccade doesn't, by definition, require a 180-turn?

5. Between subject consistency (1079ff): For the entropy metric are there enough data to make sure this measure can be made reliably? It seems like there will be many cells in the grid with very few points?

6. Figure 4: Return fixations tend to follow smaller saccades and last longer. First, given the figure caption, A and B (or the caption) could be reversed. But more critically, the duration argument is really driven by the search tasks, and it's not clear if the returns are to the to be found target so that the duration is more a function of the recognition part of the task and not really a "universal" property of return saccades?

Reviewer #2: The authors have done a good job of addressing my previous comments. The motivation, model evaluation and comparison to behavioral data are much clearer and compelling now than in the previous version

**Have the authors made all data and (if applicable) computational code underlying the findings in their manuscript fully available?**

Reviewer #1: Yes

Reviewer #2: Yes

PLOS authors have the option to publish the peer review history of their article (what does this mean?). If published, this will include your full peer review and any attached files.

Reviewer #1: No

Reviewer #2: **Yes: **Prof. Dominik Endres, PhD

Figure Files:

Data Requirements:

Reproducibility:

References:

---

## [Decision Letter · Decision Letter 2]

13 Oct 2022

Dear Prof. Kreiman,

We are pleased to inform you that your manuscript 'Look Twice: A Generalist Computational Model Predicts Return Fixations across Tasks and Species' has been provisionally accepted for publication in PLOS Computational Biology.

Best regards,

Aldo A Faisal

Academic Editor

PLOS Computational Biology

Wolfgang Einhäuser

Section Editor

PLOS Computational Biology

Jason A. Papin

Editor-in-Chief

PLOS Computational Biology

Feilim Mac Gabhann

Editor-in-Chief

PLOS Computational Biology

Reviewer's Responses to Questions

**Comments to the Authors:**

Reviewer #1: The authors have addressed my few remaining comments in this latest revision. Congratulations on an interesting addition to the literature!

Reviewer #2: thanks for the revisions, I am happy with the paper as is.

**Have the authors made all data and (if applicable) computational code underlying the findings in their manuscript fully available?**

Reviewer #1: Yes

Reviewer #2: Yes

PLOS authors have the option to publish the peer review history of their article (what does this mean?). If published, this will include your full peer review and any attached files.

Reviewer #1: No

Reviewer #2: **Yes: **Dominik M Endres

---

## [Editor Report · Acceptance letter]

7 Nov 2022

PCOMPBIOL-D-21-01364R2 

Look Twice: A Generalist Computational Model Predicts Return Fixations across Tasks and Species

Dear Dr Kreiman,

I am pleased to inform you that your manuscript has been formally accepted for publication in PLOS Computational Biology. Your manuscript is now with our production department and you will be notified of the publication date in due course.

With kind regards,

Anita Estes
